# Adolescent wellbeing is associated with positive outcomes in early adulthood in a sibling comparison study

**Anne J.M.R. Geijsen** [1,2] ✉ **& Meike Bartels** [1,2]

Puberty through adulthood involves major psychological and social change. Prospective evidence on adolescent wellbeing and later outcomes remains limited. Here we show that higher adolescent wellbeing (age 14–16) is associated with more favourable psychological and health-related outcomes in early adulthood (ages 20–25 and 25–35). Using data of 14,518 twins and siblings of the Netherlands Twin Register, we examine associations between adolescent wellbeing and adult wellbeing, flourishing, personality traits, self-rated health, and sleep quality using both between-family and within-family designs. Between-family analyses indicate positive associations with early adulthood wellbeing ($\beta = 0.32;0.38$), flourishing ($\beta = 0.27;0.34$), conscientiousness ($\beta = 0.18;0.22$), self-rated health ($\beta = 0.09;0.18$), and sleep quality ($\beta = 0.59;0.87$), and inverse associations with early adulthood neuroticism ($\beta = -0.34;-0.11$)(pFDR<0.05). Associations with self-rated health ($\beta = 0.09$), sleep quality ($\beta = 059;0.66$), and neuroticism ($\beta = -0.11;-0.37$), remain significant (pFDR<0.05) after adjusting for adolescent levels of the corresponding outcome. In the within-family analyses, that control for shared familial factors, associations with wellbeing ($\beta_{20-25} = 0.12$), flourishing ($\beta_{20-25} = 0.17$), self-rated health ($\beta_{20-25} = 0.07$), sleep quality ($\beta = 0.46;0.71$), and neuroticism ($\beta_{20-25} = -0.12$) (pFDR<0.05) were also observed. The findings characterize long-term associations between adolescent wellbeing and early adulthood outcomes, with attenuation observed after accounting for shared familial factors.

Puberty through young adulthood is a turbulent period marked by significant hormonal changes, physical growth, including rapid brain development, an increase in cognitive and intellectual capacities[1–3], and psychological and social changes[4] which can influence the wellbeing of adolescents. Despite the vulnerable nature of this period, the focus on adolescents' development often emphasizes preferred adult outcomes such as educational and income prospects[5], largely influenced by societal priorities centered on economic growth[6]. This narrow economical focus risks overlooking the quality of adolescents'

everyday lives, despite it being a window of opportunity to foster resilience, healthy lifestyles, and overall wellbeing. Understanding adolescent wellbeing may help to contextualize later outcomes across adulthood.

Wellbeing, often considered an umbrella term, encompasses various aspects of positive and negative life evaluations, emotional states, and a person's sense of meaning[7]. A conceptual distinction can be made between subjective (or hedonic) and psychological (or eudaimonic) wellbeing[8,9], where subjective wellbeing is defined as the cognitive and affective evaluation of a person's life[8,10]. Psychological

[1]Department of Biological Psychology, Faculty of Behavioural and Movement Sciences, Vrije Universiteit Amsterdam, Amsterdam, The Netherlands. [2]Amsterdam Public Health Research Institute, Amsterdam University Medical Centre, Amsterdam, The Netherlands. ✉e-mail: a.j.m.r.geijsen@vu.nl

wellbeing is summarized as positive functioning and purpose in life[9]. Many of these measures of wellbeing correlate moderately to strongly, suggesting an underlying broad wellbeing factor[11–13].

Adolescents constitute over one-sixth of the global population[14], where adolescence is the period from age 10 up to 24 years of age[15]. The 2023 Global Accelerated Action for Health of Adolescents highlights that investing in adolescent health and wellbeing yields substantial benefits, not only for individual health and wellbeing, but also in economic and social capital[16,17]. It is, furthermore, well established that mental health problems during adolescence is a risk factor for future mental problems and can lead to increased risks of later-life psychopathology[18–21]. Positive outcomes are way less studied. Most research to date studied the determinants of wellbeing[22–24] or studied the impact of wellbeing among adults on life outcomes. For example, research showed that adults with higher happiness scores live on average 14% longer compared to individuals who report to be unhappy[25,26].

Relatively few studies have been conducted investigating the role of wellbeing among adolescents and later life outcomes, while the relationship between sociodemographic, health and lifestyle-related factors and wellbeing might be reverse compared to adults or have a bidirectional nature at the least. To illustrate, among 186 adolescents of age 14, positive affect was shown to be positively associated with friendship attachment and career outcomes, and inversely associated with loneliness and anxiety at ages 23-35[27]. Furthermore, De Neve et al. showed that adolescents between 16 and 18 years of age with higher scores for wellbeing had a higher mean income at 29 years of age, while taking into account several confounding factors such as education, intelligence quotient, and physical health[28], indicating an association between higher adolescent wellbeing and more favourable later life outcomes. Importantly, in contrast to the study of Kansky et al. the latter study included sibling fixed effects, which means that the reported associations are less likely to be fully explained by stable family-level factors, although residual confounding cannot be excluded.

Studies that include multiple family members offer unique insights into the interplay between genetics and environment in shaping individual outcomes. Within-family approaches, alongside between-family analyses, allow comparison of associations before and after accounting for factors shared by siblings (for example, aspects of family environment and genetic background). Attenuation in within-family models can indicate that between-family associations may partly reflect shared familial factors; persisting within-family associations indicate differences between siblings within the same family. For instance, if a trait is more similar within a family compared to unrelated individuals, it suggests the influence of familial factors, such as shared genes and/or shared environment. Conversely, if the trait is equally similar between family members and unrelated individuals, environmental factors are likely more influential. As such, within-family approaches help to clarify relative contributions of nature and nurture[29–31].

Here, we examine associations between middle-adolescent wellbeing (ages 14–16) and psychological, health-related, and lifestyle outcomes in early adulthood (ages 20–25 and 25–35) using data from 14,518 twins and siblings from the Netherlands Twin Register. Using both between-family and within-family analyses, we assess whether associations between adolescent wellbeing and later outcomes persist after accounting for shared familial factors and earlier levels of the same outcomes. We observe statistically significant associations between adolescent wellbeing and several early adulthood outcomes, including wellbeing, flourishing, sleep quality, self-rated health, and neuroticism, with effect sizes generally attenuated

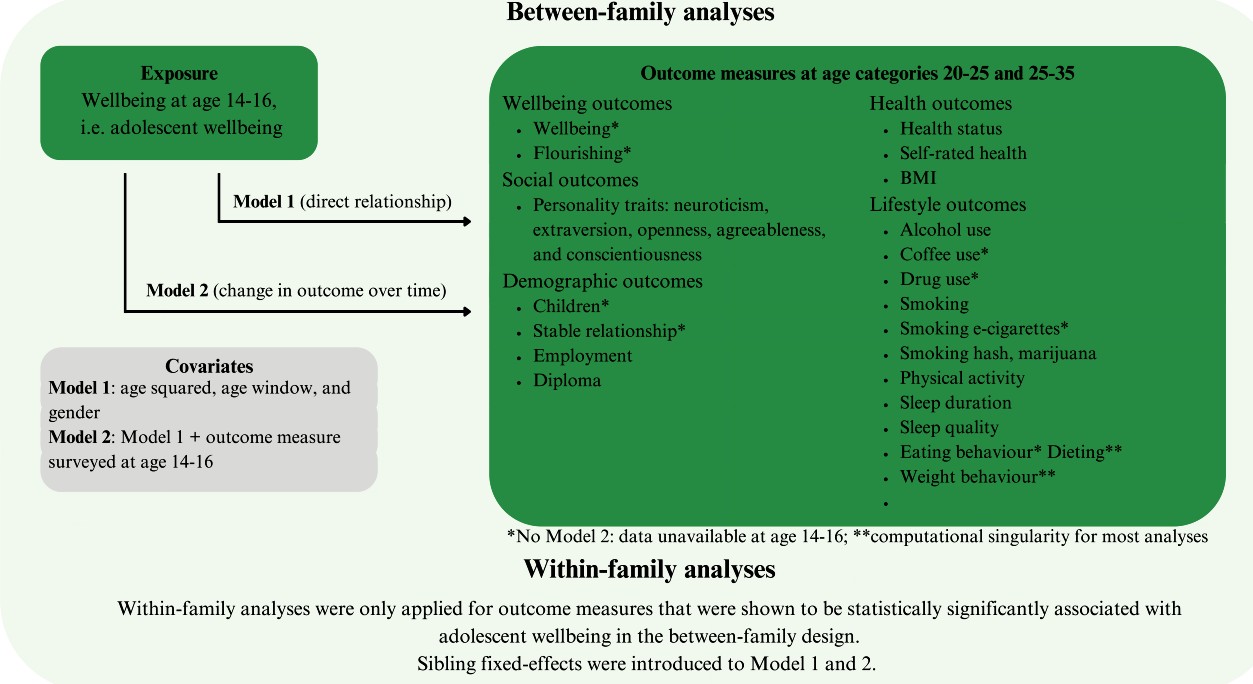

**Fig. 1 | Overview of the study design.** Adolescent wellbeing assessed at ages 14–16 was used as the exposure and several outcome measures were examined in early adulthood (ages 20–25 and 25–35). Health outcomes comprised self-rated health and BMI; lifestyle outcomes included alcohol, drug, smoking, and e-cigarettes use, physical activity, sleep, and eating-related behaviours; wellbeing outcomes included wellbeing and flourishing; social outcomes covered personality traits (neuroticism, extraversion, openness, agreeableness, conscientiousness); and demographic outcomes included children, relationship status, employment, and diploma. Two models were applied: Model 1 estimated direct associations between adolescent wellbeing and adult outcomes, adjusting for age squared, age window, and gender; Model 2 additionally adjusted for levels of the outcome at ages 14–16 to assess change over time. Analyses were conducted using both between-family and within-family (sibling fixed-effects) designs. Within-family analyses were restricted to outcomes that showed statistically significant associations in the between-family analyses. Asterisks indicate outcomes for which Model 2 could not be estimated due to unavailable baseline data (*) or computational singularity (**).

**Table 1 | Baseline and wellbeing characteristics of the study population**

| | Adolescence age (n = 14,518) | Age category 20–25 | Age category 25–35 |
|---|---|---|---|
| **Gender** | | | |
| Men | 6178 (42.6%) | | |
| Women | 8340 (57.4%) | | |
| **Age** | | | |
| Mean (SD) | 15.9 (1.8) | | |
| **Adolescence life satisfaction score** | | | |
| Mean (SD) | 27.4 (5.0) | | |
| **Adolescence subjective happiness score** | | | |
| Mean (SD) | 22.7 (4.2) | | |
| **Adolescence quality of life score** | | | |
| Mean (SD) | 7.8 (1.1) | | |
| **Adolescent wellbeing score** | | | |
| Mean (SD) | −0.00724 (0.8) | | |
| *Wellbeing outcomes* | | | |
| **Life satisfaction score** | | | |
| n | | 3799 | 1863 |
| Mean (SD) | | 27.1 (5.1) | 27.3 (5.4) |
| **Subjective happiness score**[a] | | | |
| n | | 1050 | 1472 |
| Mean (SD) | | 21.6 (4.8) | 22.2 (4.8) |
| **Quality of life score** | | | |
| n | | 3808 | 1871 |
| Mean (SD) | | 7.5 (1.1) | 7.6 (1.1) |
| **Wellbeing score** | | | |
| n | | 3810 | 1872 |
| Mean (SD) | | −0.00892 (0.8) | 0.0478 (0.9) |
| **Flourishing score** | | | |
| n | | 1364 | 1649 |
| Mean (SD) | | 45.9 (6.2) | 46.3 (5.9) |

Numbers are presented as n(%) or Mean (SD), unless specified otherwise. <u>Note</u>: Not all participants received all surveys, and not all outcome measures were included in every survey. To maximize the time window between adolescent wellbeing and the outcome measure, the survey that provided the longest interval between the two was selected. As a result, the number of individuals included per outcome varies. Please review the Methods section for all details on the questionnaires and measures used.
[a]The subjective happiness scale is not available in all adult questionnaires, therefore sample sizes are lower compared to the other wellbeing measures.

in within-family models. These results describe the long-term predictive associations of adolescent wellbeing across multiple outcome domains.

## Results

Key methodological details are summarized below, with full descriptions provided in the Methods section. An overview of the study design is provided in Fig. 1. Characteristics of the study population are described in Tables 1–4. The mean (range) scores on individual adolescent wellbeing measures were 27.4 (5–35), 22.7 (4–28), and 7.8 (1–10) for life satisfaction, subjective happiness, and quality of life, respectively.

### Between-family regression models

Results of the between-family regression models for age categories 20–25 and 25–35 are shown in Figs. 2 and 3, respectively. Forest plots of statistically significantly associations between outcome measures and adolescent wellbeing are presented in Figs. 4 and 5. Detailed information on all regression model results for Model 1 and Model 2 are presented in Supplementary Tables S3–6.

**Table 2 | Social, health and demographics outcome characteristics of the study population**

| | Age category 20–25 | Age category 25–35 |
|---|---|---|
| *Social outcomes* | | |
| **Personality - Neuroticism** | | |
| n | 2923 | 570 |
| Mean (SD) | 31.6 (8.2) | 31.4 (8.6) |
| **Personality - Extraversion** | | |
| n | 2923 | 570 |
| Mean (SD) | 43.0 (6.2) | 42.4 (6.4) |
| **Personality - Openness** | | |
| n | 2923 | 570 |
| Mean (SD) | 38.5 (6.4) | 39.0 (6.5) |
| **Personality - Agreeableness** | | |
| n | 2923 | 570 |
| Mean (SD) | 44.0 (5.6) | 44.3 (5.4) |
| **Personality - Conscientiousness** | | |
| n | 2923 | 570 |
| Mean (SD) | 45.1 (6.2) | 45.6 (6.1) |
| *Demographic outcomes* | | |
| **Children** | | |
| No | 3948 (98.1%) | 1691 (83.0%) |
| Yes | 78 (1.9%) | 347 (17.0%) |
| **Stable relationship** | | |
| No | 2726 (66.7%) | 653 (32.1%) |
| Yes | 1361 (33.3%) | 1384 (67.9%) |
| **Employment** | | |
| No | 1650 (42.2%) | 903 (52.0%) |
| Yes | 2258 (57.8%) | 835 (48.0%) |
| **Diploma** | | |
| No | 1420 (27.5%) | 297 (12.0%) |
| Yes | 3737 (72.5%) | 2186 (88.0%) |
| *Health outcomes* | | |
| **Health status (illnesses)** | | |
| No | 400 (65.6%) | 81 (46.0%) |
| Yes | 210 (34.4%) | 95 (54.0%) |
| **Self-rated health** | | |
| n | 5161 | 2349 |
| Mean (SD) | 4.2 (0.7) | 4.1 (0.7) |
| **BMI (kg/m²)** | | |
| n | 5859 | 2664 |
| Mean (SD) | 22.7 (3.4) | 23.4 (3.8) |

Numbers are presented as n(%) or Mean (SD), unless specified otherwise. <u>Note</u>: Not all participants received all surveys, and not all outcome measures were included in every survey. To maximize the time window between adolescent wellbeing and the outcome measure, the survey that provided the longest interval between the two was selected. As a result, the number of individuals included per outcome varies. Please review the Methods section for all details on the questionnaires and measures used.

The following outcome measures at age 20–25 and 25–35 did not show statistically significant associations with adolescent wellbeing in Model 1 or Model 2: demographic outcomes, health status, alcohol use, drug use, and sleep duration (Figs. 4 and 5). As a result of computational singularity, Model 1 for age category 25–35 and Model 2 for both age categories were not calculated for the outcome measures dieting and weight behaviour.

**Wellbeing outcomes.** Wellbeing and flourishing at age 20–25 and 25–35 were positively statistically significantly associated with adolescent wellbeing in Model 1 with pFDR-values < 0.05 (Figs. 4, 5 and

## Table 3 | Lifestyle outcome characteristics of the study population (part 1)

| Lifestyle outcomes | Age category 20–25 | Age category 25–35 |
|---|---|---|
| **Alcohol use** | | |
| less than 1 glass a day | 113 (6.7%) | 28 (8.4%) |
| 1–2 glasses a day | 312 (18.5%) | 59 (17.7%) |
| 3–5 glasses a day | 389 (23.0%) | 95 (28.4%) |
| 6–10 glasses a day | 441 (26.1%) | 92 (27.5%) |
| 11–20 glasses a day | 305 (18.1%) | 50 (15.0%) |
| 21–40 glasses a day | 115 (6.8%) | 9 (2.7%) |
| more than 40 glasses a day | 14 (0.8%) | 1 (0.3%) |
| **Coffee use (cups/day)** | | |
| n | 868 | 237 |
| Mean (SD) | 3.01 (1.8) | 3.07 (1.6) |
| **Drug use** | | |
| No | 2541 (87.5%) | 491 (85.8%) |
| Yes | 364 (12.5%) | 81 (14.2%) |
| **Smoking** | | |
| Never | 3268 (56.6%) | 1797 (66.0%) |
| In the past | 864 (15.0%) | 458 (16.8%) |
| Current | 1642 (28.4%) | 468 (17.2%) |
| **Smoking e-cigarettes** | | |
| No | 2219 (91.6%) | 506 (92.2%) |
| Yes | 203 (8.4%) | 43 (7.8%) |
| **Smoking hash, marijuana** | | |
| No | 1761 (65.1%) | 1144 (64.9%) |
| Yes | 943 (34.9%) | 618 (35.1%) |
| **Physical activity** | | |
| Sedentary | 1777 (39.0%) | 1002 (47.1%) |
| Moderate | 1875 (41.1%) | 769 (36.2%) |
| Vigorous | 906 (19.9%) | 356 (16.7%) |

Numbers are presented as n(%) or Mean (SD), unless specified otherwise. Note: Not all participants received all surveys, and not all outcome measures were included in every survey. To maximize the time window between adolescent wellbeing and the outcome measure, the survey that provided the longest interval between the two was selected. As a result, the number of individuals included per outcome varies. Please review the Methods section for all details on the questionnaires and measures used.

## Table 4 | Lifestyle outcome characteristics of the study population (part 2)

| Lifestyle outcomes | Age category 20–25 | Age category 25–35 |
|---|---|---|
| **Sleep duration (h)** | | |
| N | 554 | 29 |
| Mean (SD) | 8.8 (1.2) | 8.3 (0.8) |
| **Sleep quality** | | |
| No trouble sleeping | 2555 (73.4%) | 1146 (69.0%) |
| Some trouble sleeping | 678 (19.5%) | 365 (22.0%) |
| Often trouble sleeping | 250 (7.2%) | 150 (9.0%) |
| **Eating behaviour** | | |
| Stops eating before full | 398 (17.8%) | 88 (22.6%) |
| Stops eating when full | 1612 (72.2%) | 267 (68.5%) |
| Continues eating when full | 222 (9.9%) | 35 (9.0%) |
| **Dieting** | | |
| Never went on a diet | 684 (69.2%) | 24 (48.0%) |
| Went on a diet a couple of times | 208 (21.0%) | 18 (36.0%) |
| Went on a diet several times | 49 (5.0%) | 5 (10.0%) |
| Often on a diet | 33 (3.3%) | 2 (4.0%) |
| Always on a diet | 15 (1.5%) | 1 (2.0%) |
| **Weight behaviour** | | |
| Not afraid to gain weight | 377 (38.1%) | 16 (32.0%) |
| Somewhat afraid to gain weight | 378 (38.2%) | 14 (28.0%) |
| Quite afraid to gain weight | 141 (14.2%) | 14 (28.0%) |
| Very afraid to gain weight | 78 (7.9%) | 4 (8.0%) |
| Extremely afraid to gain weight | 16 (1.6%) | 2 (4.0%) |

Numbers are presented as n(%) or Mean (SD), unless specified otherwise. Note: Not all participants received all surveys, and not all outcome measures were included in every survey. To maximize the time window between adolescent wellbeing and the outcome measure, the survey that provided the longest interval between the two was selected. As a result, the number of individuals included per outcome varies. Please review the Methods section for all details on the questionnaires and measures used.

Supplementary Tables S3–6). Higher adolescent wellbeing was associated with higher wellbeing scores at age 20–25 and 25–35, with $z(1) = 19.76$, standardized beta $(B_{st}) = 0.38$ (95% CI: 0.34; 0.42, $p = 0.000$, incremental pseudo $R^2$ $(R_{incr})$: 13.7%) and $z(1) = 12.55$ and $B_{st} = 0.38$ (95% CI: 0.27; 0.37, $p = 0.000$, $R_{incr}$: 10.1%) at age 20–25 and 25–35, respectively. Adolescent wellbeing was positively associated with flourishing scores at age 20–25 with a $z(1)$ of 10.62 and $B_{st}$ of 0.34 (95% CI: 0.28; 0.41, $p = 0.000$, $R_{incr}$: 11.4%) and at age 25–35 with a $z(1)$ of 8.99 and a $B_{st}$ of 0.27 (95% CI: 0.21; 0.33, $p = 0.000$, $R_{incr}$: 6.9%).

**Social outcomes.** Neuroticism in early adulthood showed a statistically significant inverse relationship with adolescent wellbeing at age 20–25 and 25–35 with a $z(1)$ of −17.59 and a $B_{st}$ of −0.34 (95% CI: −0.38; −0.30, $p = 0.000$, $R_{incr}$: 10.9%) and a $z(1)$ of −9.09 and a $B_{st}$ of −0.34 (95% CI: −0.42; −0.27, $p = 0.000$, $R_{incr}$: 12.6%), respectively in Model 1. The inverse relationships remained statistically significant after additionally adjusting for adolescence neuroticism scores for both age categories in Model 2, see Figs. 4 and 5, with the association remaining statistically significant after adjustment for adolescent neuroticism.

Openness at age 20–25 was statistically significantly inversely associated with adolescent wellbeing with a $B_{st}$ −0.05 (95% CI: −0.09;

−0.01, $R_{incr}$: 0.2%), but not at age 25–35. The association at age 20–25 did not remain statistically significant when additionally adjusting for adolescence openness scores. The personality traits extraversion and conscientiousness showed a statistically significant positive association with adolescent wellbeing at age 20–25 and 25–35 in Model 1. When additionally adjusting for adolescence conscientiousness scores in Model 2, conscientiousness remained statistically significantly associated with adolescent wellbeing with a $B_{st}$ of 0.08 (95% CI: 0.03;0.13, $R_{incr}$: 0.2%) at age 20–25, but not at age 25–35. Extraversion was not statistically significantly associated with adolescent wellbeing in Model 2.

Agreeableness at age 20–25 and age 25–35 showed a positive statistically significant relationship with adolescent wellbeing with a $B_{st}$ of 0.09 (95% CI: 0.05; 0.13, $R_{incr}$: 0.8%) and $B_{st}$ of 0.10 (95% CI: 0.02; 0.17, $R_{incr}$: 0.8%), respectively, in Model 1, but not in Model 2 when additionally adjusting for adolescence agreeableness.

**Health outcomes.** Self-rated health at age 20–25 and 25–35 showed positive statistically significant associations with adolescent wellbeing in Model 1 with a $z(1)$ of 12.21 and a $B_{st}$ of 0.18 (95% CI: 0.15; 0.21, $p = 0.000$, $R_{incr}$: 3.2%) and a $z(1)$ of 7.81 and a $B_{st}$ of 0.18 (95% CI: 0.13; 0.22, $p = 0.000$, $R_{incr}$: 3.1%), respectively. The relationship remained statistically significant after additionally adjusting for adolescence self-rated health score at age 20–25 ($B_{st}$: 0.09 (95% CI: 0.06; 0.12, $R_{incr}$: 0.7%)) and 25–35 ($B_{st}$: 0.09 (95% CI: 0.04; 0.13, $R_{incr}$: 0.7%)).

**Fig. 2 | Between-family associations between adolescent wellbeing and outcome measurs (age 20–25).** Between-family regression models examining associations between adolescent wellbeing and wellbeing (magenta), social (yellow), demographic (brown), health (green), and lifestyle (blue) outcomes are shown. Points represent standardised regression coefficients (β). The vertical green line indicates the null effect (β = 0). Positive and inverse associations with adolescent wellbeing are shown to the right and left of the line, respectively. The y-axis shows FDR-adjusted p values on a logarithmic scale; the horizontal red dotted line indicates the significance threshold (pFDR = 0.05), based on the Benjamini–Hochberg procedure. All tests were two-sided. Exact p values and full model results are provided in Supplementary Tables S3–6.

A similar pattern emerged for BMI, which showed small but statistically significant inverse associations with adolescent wellbeing in Model 1: higher wellbeing was associated with lower BMI at age 20–25 ($z(1)$: −4.81, $B_{st}$: −0.07 (95% CI: −0.13; −0.01, $p = 0.000$, $R_{incr}$: 0.5%)) and 25–35 ($z(1)$: −3.59, $B_{st}$: −0.08 (95% CI: −0.12; −0.04, $p = 0.000$, $R_{incr}$: 0.8%)). When additionally adjusting for adolescent BMI in Model 2, BMI remained inversely associated with adolescent wellbeing with a $B_{st}$ of −0.04 (95% CI: −0.06; −0.01, $R_{incr}$: 0.1%) at age 20–25, but not at age 25–35. (Figs. 4 and 5).

**Lifestyle outcomes.** Coffee use at age 20–25, but not at age 25–35, had a statistically significant inverse association with adolescent wellbeing in Model 1 ($B_{st}$ of −0.07 (95% CI: −0.13; −0.01, $R_{incr}$: 0.5%)).

Participants who never smoked at age 20-25 showed a positive statistically significant association with adolescent wellbeing in Model 1 ($B_{st}$ of 0.10 (95% CI: 0.03; 0.17)) compared to participants who were current smokers at 20–25. This relationship was not observed when comparing participants who smoked in the past compared to current smokers at age 20–25. No statistically significant association was observed when additionally adjusting for adolescence smoking habits (Model 2). Adolescent wellbeing was not associated with smoking habits at age 25–35, nor with use of e-cigarettes at age 20-25 or 25–35.

In contrast, smoking hash or marijuana at age 20–25 and at age 25–35, showed inverse statistically significant associations with adolescent wellbeing in Model 1 ($B_{st}$ of −0.33 (95% CI: −0.50; −0.16, $R_{incr}$: 0.5%) and $B_{st}$ of −0.36 (95% CI: −0.57; −0.14, $R_{incr}$: 0.6%), respectively).

The association remained statistically significant after additionally adjusting for adolescence smoking of hash or marijuana in Model 2 at age 20-25, but not at age 25–35.

In Model 1 and 2, participants with a sedentary lifestyle at age 20–25 and 25–35 showed an inverse statistically significant association with adolescent wellbeing with a $B_{st}$ of −0.18 (95% CI: −0.30;−0.06) and a $B_{st}$ of −0.34 (95% CI: −0.56;−0.13), respectively, for Model 2, when compared to participants who are vigorously active.

Finally, positive associations were observed between adolescent wellbeing and sleep quality in early adulthood. Participants who report to have no trouble sleeping at age 20−25 and 25−35 showed a positive statistically significant association with adolescent wellbeing with a $B_{st}$ of 0.87 (95% CI: 0.68; 1.05) and a $B_{st}$ of 0.85 (95% CI: 0.63; 1.08), respectively, when compared to participants who report often to have trouble sleeping in Model 1. The positive associations between adolescent wellbeing and sleep patterns remained statistically significant in Model 2 (see Figs. 4 and 5).

**Within-family regression models**
Within-family results are shown in Figs. 4 and 5 for outcome measures that were statistically significantly associated with adolescent wellbeing in the between-family analyses. Detailed information on all regression model results for Model 1 and Model 2 are presented in Supplementary Table S3–6. Within-family regression models account for factors shared by siblings within a family, allowing estimation of associations with differences between siblings. Effect estimates from within-family models can be compared with between-family

**Age category 25-35 (Model 1 and 2)**

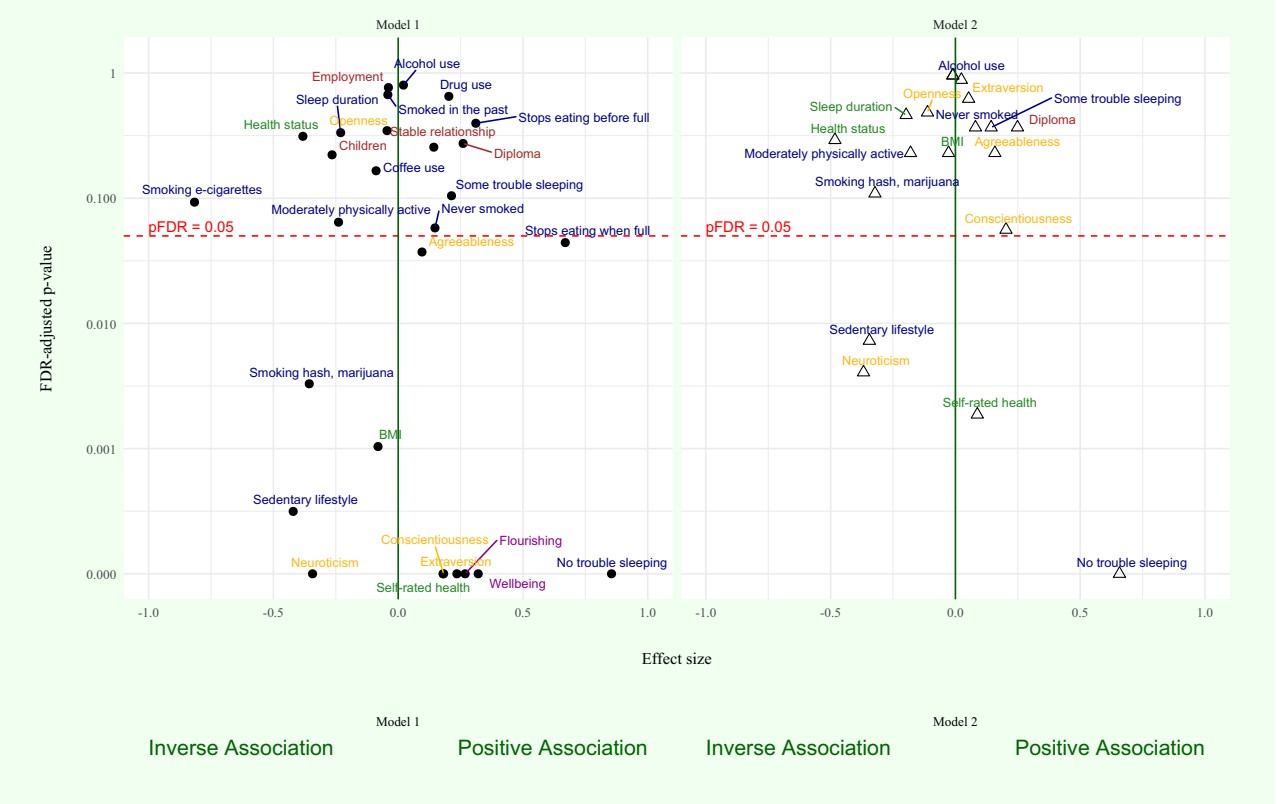

**Fig. 3 | Between-family associations between adolescent wellbeing and outcome measures (age 25–35).** Between-family regression models examining associations between adolescent wellbeing and wellbeing (magenta), social (yellow), demographic (brown), health (green), and lifestyle (blue) outcomes are shown. Points represent standardised regression coefficients (β). The vertical green line indicates the null effect (β = 0). Positive and inverse associations with adolescent wellbeing are shown to the right and left of the line, respectively. The y-axis shows FDR-adjusted p values on a logarithmic scale; the horizontal red dotted line indicates the significance threshold (pFDR = 0.05), based on the Benjamini–Hochberg procedure. All tests were two-sided. Exact p values and full model results are provided in Supplementary Tables S3–6.

estimates to assess attenuation after accounting for shared familial factors.

Wellbeing and flourishing at age 20-25 were associated with adolescent wellbeing with a $B_{st}$ of 0.12 (95% CI: 0.06;0.19) and $B_{st}$ of 0.17 (95% CI: 0.06; 0.28), respectively, but not age 25–35. Associations were not statistically significant at ages 25–35 after correction for multiple testing; sample sizes were smaller in this age category.

The personality trait neuroticism at age 20–25, but not age 25–35, showed an inverse statistically significant relationship with adolescent wellbeing with a $B_{st}$ of −0.12 (95% CI: −0.19; −0.05) in Model 1. When additionally adjusting for adolescence neuroticism scores, the inverse relation with adolescent wellbeing became non-significant in Model 2. Reduced sample sizes in Model 2 and at later ages may have limited the ability to detect effects of comparable magnitude.

Similarly, self-rated health at age 20-25, but not age 25–35, showed a positive statistically significant association with adolescent wellbeing in Model 1. When additionally adjusting for adolescence self-rated health, the association did not remain statistically significant in Model 2.

Participants who report to have no trouble sleeping at age 20–25 and 25–35 showed a positive statistically significant association with adolescent wellbeing with a $B_{st}$ of 0.46 (95% CI: 0.17; 0.75) and a $B_{st}$ of 0.71 (95% CI: 0.28; 1.13), respectively, when compared to participants who report often to have trouble sleeping in Model 1. After adjustment for adolescent sleep quality, the associations were attenuated and were not statistically significant (Model 2).

## Discussion

In the current study, the associations between middle-adolescent wellbeing (age 14–16) and sociodemographic, health and lifestyle outcomes in early adulthood (age 20–25 and 25–35) were investigated using both between-family and within-family designs among 14,518 twins and their siblings of the Netherlands Twin Register. In line with our preregistered expectations, several sociodemographic, health, and lifestyle outcomes were associated with adolescent wellbeing. However, not all outcomes showed significant associations, and the direction of effects varied across outcome measures. Between-family analyses showed positive associations between adolescent wellbeing and wellbeing, flourishing, conscientiousness, self-rated health and sleep quality in early adulthood, and inverse associations with early adulthood neuroticism and smoking habits. Notably, several associations between adolescent wellbeing and early adulthood outcomes remained statistically significant after adjusting for adolescent levels of these outcomes. The associations with wellbeing, flourishing, self-rated health, sleep quality and neuroticism were also observed when including a sibling fixed effect.

An earlier study investigated the relationship between adolescent wellbeing and later life outcomes. More specifically, the association between adolescent wellbeing and mean income at age 29 was studied, indicating that higher wellbeing among adolescents was associated with more positive later life outcomes[28]. Mean income was not available in the present cohort making it impossible to compare results. However, the current results partly support the hypothesis that higher

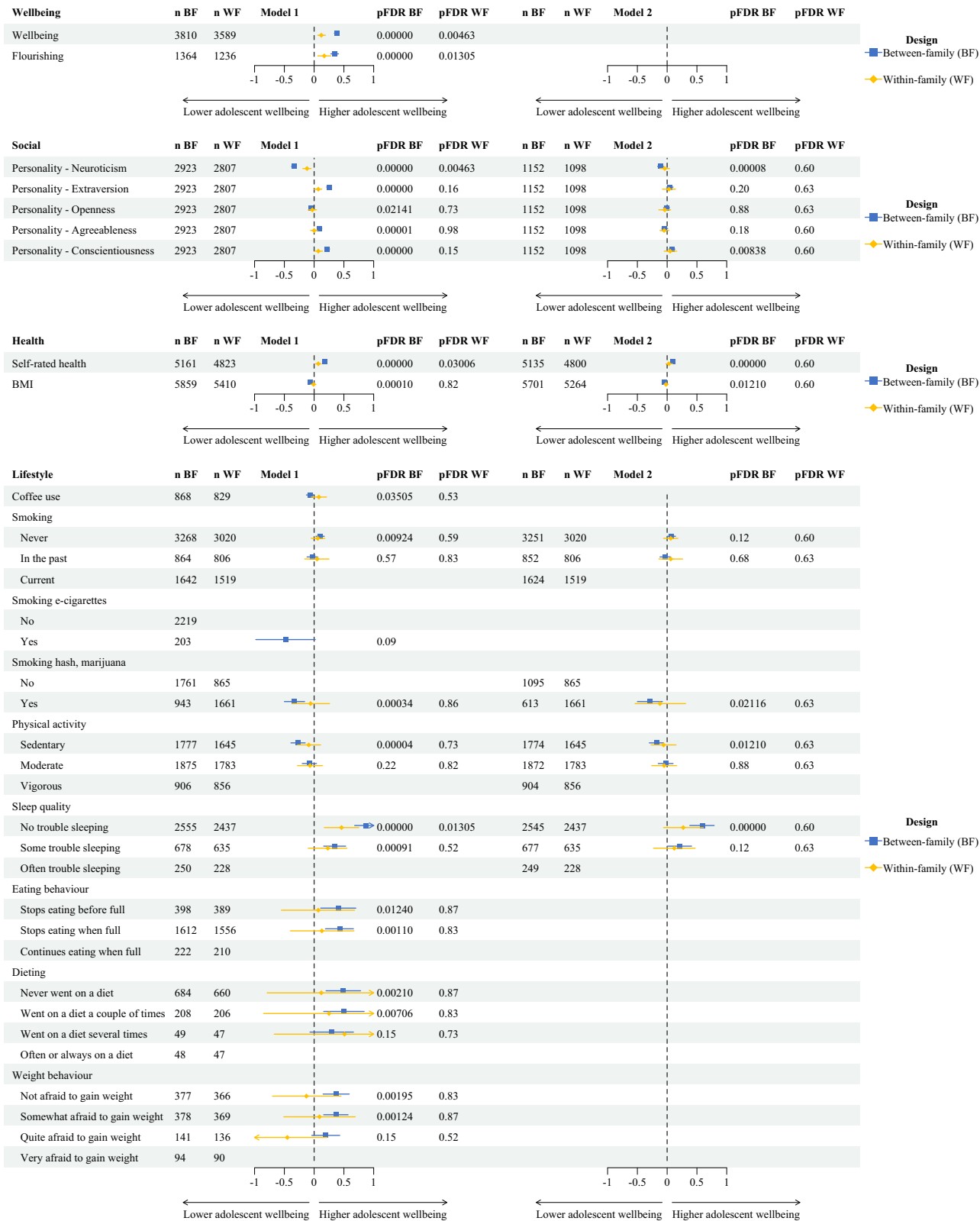

**Fig. 4 | Forest plots for between-family and within-family regression models (Models 1 and 2) for age category 20–25.** Outcome measures associated with adolescent wellbeing are shown. Blue and yellow symbols represent between-family (BF) and within-family (WF) regression analyses, respectively. Points indicate standardized regression coefficients (β), and horizontal lines indicate 95% confidence intervals. The vertical dotted line represents a null effect (β = 0). Positive and inverse associations with adolescent wellbeing are shown on the right and left sides of the line, respectively. All associations were estimated using GEE regression models with two-sided tests; p values were adjusted for multiple testing using the Benjamini–Hochberg false discovery rate (FDR) procedure. Exact FDR-adjusted p values are shown alongside the estimates. Detailed regression results for Models 1 and 2 are provided in Supplementary Tables S3–6.

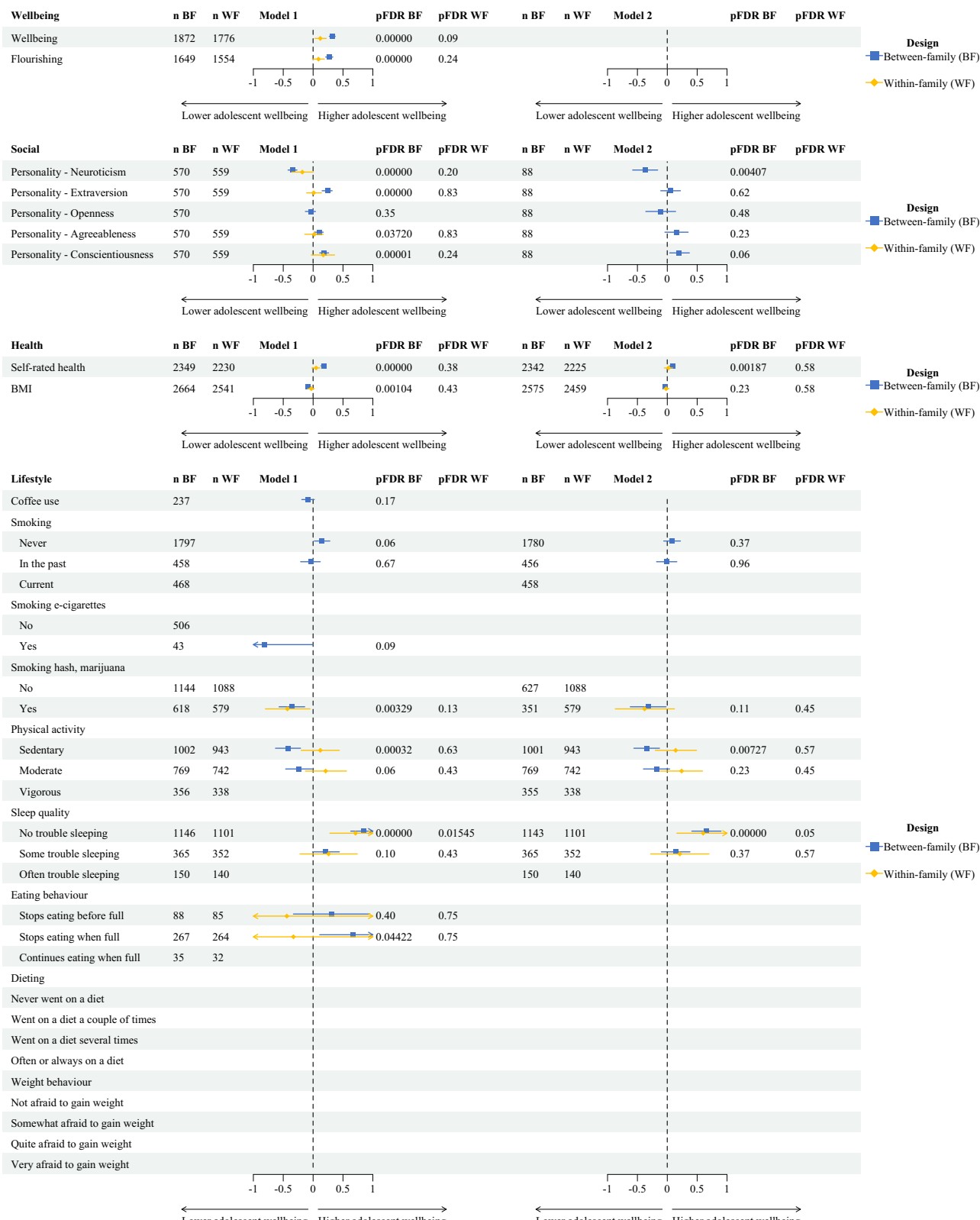

**Fig. 5 | Forest plots for between-family and within-family regression models (Models 1 and 2) for age category 25–35.** Outcome measures associated with adolescent wellbeing are shown. Blue and yellow symbols represent between-family (BF) and within-family (WF) regression analyses, respectively. Points indicate standardized regression coefficients (β), and horizontal lines indicate 95% confidence intervals. The vertical dotted line represents a null effect (β = 0). Positive and inverse associations with adolescent wellbeing are shown on the right and left sides of the line, respectively. All associations were estimated using GEE regression models with two-sided tests; p values were adjusted for multiple testing using the Benjamini–Hochberg false discovery rate (FDR) procedure. Exact FDR-adjusted p values are shown alongside the estimates. Detailed regression results for Models 1 and 2 are provided in Supplementary Tables S3–6.

adolescent wellbeing is associated with more positive outcomes in early adulthood.

## Between-family analyses

In the current study, between-family analyses revealed several associations between adolescent wellbeing and outcomes in early adulthood. In the between-family models, positive associations between adolescent wellbeing and wellbeing, flourishing, self-rated health and sleep quality in early adulthood were observed. The positive relationship between adolescent wellbeing and self-rated health and sleep quality in early adulthood remained when adjusting for adolescence self-rated health and adolescence sleep quality, respectively. Similarly, an inverse relationship between adolescent wellbeing and later life neuroticism was observed when taking into account adolescence neuroticism. Adolescent flourishing was not available, therefore, Model 2 could not be tested. These observations are consistent with individual differences in wellbeing, indicating that wellbeing is a flexible phenotype beyond adolescence. The plasticity of wellbeing has been suggested before[32–34] and underscores the potential for intervention protocols aimed at improving wellbeing during adolescence, a critical period with long-term implications for adult outcomes.

Statistically significant associations between adolescent wellbeing and early adulthood outcomes such as extraversion, openness, agreeableness, (e-)cigarettes, hash and marijuana smoking habits, and physical activity did not remain statistically significant when adjusting for the respective outcome measure at age 14–16. Personality traits have been shown to be relatively stable over time[35], therefore, it is to be expected that modelling the change in these specific outcome measures over time did not result in statistically significant findings.

The absence of statistical significant associations for smoking habits and physical activity could suggest that, on the one hand, environmental factors, such as influences prior to adolescence, may have a great impact and a foundation is set by the time children reach adolescence age. Another explanation can be that certain outcomes in early adulthood are largely driven by genetic factors, which is shown by the lack of a statistically significant relationship in the within-family design compared to the between-family design. Reasonably, a combination of the latter two explanations is highly sensible, where an interplay between genetics and environmental influences explain the results. Physical activity, for example, has been shown to have a genetic component[36–38], in addition to research suggesting that exposure to physical activity during childhood promotes positive later life outcomes[39–41].

The lack of statistical associations between adolescent wellbeing and early adulthood outcomes, such as alcohol and drug use, in between-family analyses are interesting in themselves. The latter results indicate that adolescent wellbeing does not necessarily have to define how adolescents behave in early adulthood. It demonstrates the need for research into the influence of life experiences prior to adolescence age on adolescence wellbeing, adolescence sociodemographic, health, and lifestyle factors and the relationship with later life outcomes.

## Within-family analyses

In addition to between-family analyses, within-family analyses using siblings were performed in the current study for the statistically significant associations between adolescent wellbeing and early adulthood outcomes in the between-family models. Within-family designs using siblings provide a powerful framework to disentangle potential genetic and or environmental influences[31,42].

Adolescent wellbeing showed a positive relationship with wellbeing, flourishing, and sleep quality in early adulthood when correcting for effects of the family environment by applying within-family models. The effect estimates of the within-family analyses were smaller compared to the between-family analyses, implying gene-environment correlations might be present[31,43].

Outcome measures such as self-rated health, BMI, physical activity and smoking and eating habits did not show statistically significant associations with adolescent wellbeing in the within-family analyses. This may suggest that it is less likely that there is a direct causal relationship between adolescent wellbeing and these outcome measures in early adulthood. Moreover, the significant between-family results may be partly due to shared environments, passive gene-environment correlations, and/or confounding factors that are shared within families but differ between families, such as population stratification, assertive mating, and/or environmental factors related to genetics[31,44–46].

## Disentangling prediction from causation

Furthermore, the distinction between prediction and causality is critical. Our findings show that adolescent wellbeing is consistently associated with multiple adult outcomes, with associations remaining robust over a follow-up period of 10–15 years, underscoring its predictive utility. When controlling for baseline levels and shared familial factors via within-family analyses, the number of significant associations was reduced, suggesting that some associations may reflect confounding by stable family-level influences or pre-existing differences. However, several associations, most notably with wellbeing, flourishing, and sleep quality at age 20–25, remained statistically significant even after these stringent adjustments. These findings indicate that several associations persist after accounting for shared familial factors and earlier levels of the outcomes and highlight its value for early identification and preventive interventions. Future work integrating predictive and causal modelling could help clarify when and how wellbeing exerts long-term influence on life trajectories.

## Limitations and strengths

During the course of the analysis, challenges were encountered related to the computational stability of certain statistical models. Specifically, attempts to fit a multinomial regression using the specified predictors resulted in a computational singularity, likely due to a combination of factors, including the limited sample size, low variability in some variables, and the inclusion of multiple predictors. The limited sample size mainly applied to the later cohort of the age category 25–35, where sample sizes were considerably lower compared to the age category 20–25, which may explain why some outcomes are statistically significant between adolescent wellbeing and later life outcome at age 20–25, but not 25–35.

Consequently, some intended analyses could not be performed. This limitation underscores the need for larger sample sizes and the careful selection of predictors in future studies to ensure sufficient variability and statistical power to support more complex modelling approaches. Importantly, variation in outcome-specific sample sizes likely reduced statistical power for certain analyses. For example, sleep duration at ages 25–35 was based on a relatively small subsample ($n = 29$). Results based on such limited sample sizes should be interpreted with caution. Further studies with larger and more diverse populations, especially among non-Western and lower-income populations that are currently underrepresented in literature[47], may help to address these issues and allow for more robust modelling of the relationships of interest. Beyond sample size considerations, notable differences emerged between the 20–25 and 25–35 age groups. For example, adolescent wellbeing predicted conscientiousness and cannabis use in the younger group but not in the older group. One possible explanation is that early adulthood (age 20–25) is a period characterized by major life transitions such as entering higher[48,49] education, establishing independence from parents, forming new peer and romantic relationships, and experimenting with health behaviours. By contrast, later young adulthood (age 25–35) often involves greater stability in work, family, and lifestyle patterns, as well as

increased health and caregiving responsibilities[47]. These developmental and environmental shifts may alter the extent to which adolescent wellbeing predicts later outcomes, potentially explaining the observed age-dependent differences.

Further, it is important to note that ages 14–16 correspond to middle adolescence, and wellbeing and its correlates may vary across different developmental stages[47] highlighting that some influences on wellbeing at this age may have originated earlier in childhood or adolescence. These findings highlight the importance of further exploring age-specific mechanisms in future research, including life exposures and outcomes before, during adolescence and beyond age 35, to study in future studies to explore the relationship whether adolescent wellbeing and life outcomes in later adulthood.

A major strength of the present study in the longitudinal design of the used NTR cohort, which resulted in the availability of several adolescence measures for the identical early adulthood outcomes. Although not available for all outcomes, Model 2, in which analyses were additionally adjusted for the outcome measure at age 14–16, added invaluable information on the potential catalysing effect of adolescent wellbeing on early adulthood outcomes since it reflects the change in a particular outcome measure. It highlighted the complex relationship of wellbeing across life phases and the importance of continuing to collect longitudinal data from birth until late age using both between-family and within-family designs. Applying both between-family and within-family models allows comparison of associations before and after accounting for shared familial factors. Although our within-family design adjusts for shared familial influences, including both genetic and shared environmental factors, it does not allow us to decompose these influences separately. Future studies could apply classical twin models (e.g., ACE modelling) to estimate the proportion of variance attributable to additive genetic, shared environmental, and non-shared environmental factors, particularly for those associations that remained robust in the within-family analyses.

In summary, adolescent wellbeing was statistically significantly associated with multiple outcomes in early adulthood in between-family analyses, with attenuation observed in within-family analyses. Several associations remained statistically significant after adjustment for shared familial factors and earlier levels of the outcomes. These findings describe long-term associations between adolescent wellbeing and early adulthood outcomes in a large longitudinal cohort. While the reported findings are promising, they also underscore the need for continued research in this area, particularly with larger and more diverse populations, to further elucidate the complex relationships between adolescent wellbeing and later life outcomes.

## Methods

All analyses were preregistered on April 17th, 2024, see https://doi.org/10.17605/OSF.IO/7DW8Z). Minor deviations from the preregistrations are explained in the subsequent subsections.

### Study design

Multiple waves of the longitudinal database of the Netherlands Twin Register (NTR) were used with data collected through self-reported questionnaires at various time points since 1987. NTR is a population-based sample of twins and their families who register voluntarily to participate. Detailed information on NTR is described in earlier reports[48–51]. NTR may be seen as a good representation of the majority of the Dutch population, although females and higher educated individuals are somewhat overrepresented and non-Caucasian individuals are underrepresented[52]. All participants in the NTR provided informed consent for participation and data use in research, in accordance with Dutch ethical guidelines. Participants did not receive financial compensation for participation in the present study.

### Participants and questionnaires

We used adolescent data collected by the NTR around ages 14–16, with a focus on participants with available wellbeing and outcome measures. The sample size to assess adolescence wellbeing, using three wellbeing measures, resulted in 14,518 adolescents with available adolescent wellbeing measures. Characteristics of the study population are described in Tables 1–4. About 58% of the study population are women. The mean (SD) age of the adolescents was 15.9 (1.8) years. Sex was recorded in the NTR based on participant self-report and coded as male or female. No data on gender identity were used in the present dataset. Sex was included as a covariate in all statistical models. The present study was not designed to test sex-specific effects; therefore, analyses were not stratified by sex. Questionnaires were distributed in waves to adolescent twins and their siblings, following parental consent. Because of the timing of consent procedures and data collection, the average age of respondents was slightly higher than the target range, with a mean of 15.9 years.

### Ethics approval

Participants in the Netherlands Twin Register (NTR) voluntarily enrol in an ongoing longitudinal research programme. Ethical approval has been obtained for NTR registration and for the collection, storage, and use of data for scientific research.

The analyses reported in the current manuscript are secondary analyses of existing NTR data. According to Dutch regulations, such secondary analyses do not fall under the Wet medisch-wetenschappelijk onderzoek met mensen (WMO), as no new data are collected and participants are not subjected to procedures or rules of behaviour. Consequently, additional approval from a local Medical Ethics Review Committee (IRB/METC) was not required.

For each survey wave, participants provided informed consent for the use of their data for research purposes and for linkage with previously collected NTR data. Only participants who consented to record linkage were included. Ethical approval numbers for the NTR data collections include YNTR14/YNTR16 (2003/182; 2010; 2012; 2013), ANTR8 (NL25220.029.08/2008-244), ANTR10 (2011/334; 2012/433), ANTR14 (2018/389; VCWE-2018-124), and genetic data (04.001.98).

### Adolescent wellbeing at age 14–16

Life satisfaction was measured by the Satisfaction with Life Scale (SWLS)[10]. The SWLS questionnaire consists of five questions that can be answered on a seven-point scale ranging from 'strongly disagree' (i.e. score 1) to 'strongly agree' (i.e. score 7). As a result, the life satisfaction score can range from 1 to 35.

Subjective happiness was determined using the scores on the Subjective Happiness Scale (SHS)[53]. The SHS questionnaire consists of four statements rated on a seven-point scale ranging from 'strongly disagree' (i.e. score 1) to 'strongly agree' (i.e. score 7); the subjective happiness score can range from 1 to 28.

Lastly, quality of life is defined by the score on the Cantril Ladder (CL)[54]. The CL consists of a 10-step ladder on which the participant can indicate where they place their lives in general, ranging from 'the worst possible life' (i.e. step 1) to 'the best possible life' (i.e. step 10).

Adolescent wellbeing was constructed as a single score using a latent score based upon the individual wellbeing measures life satisfaction, subjective happiness, and quality of life[55]. This approach follows prior research demonstrating that integrating multiple related indicators captures a broader, more reliable construct of wellbeing[56]. These measures are reliable and valid in adolescent populations and capture complementary aspects of wellbeing[57]. In our sample, reliabilities (Cronbach's alpha) were 0.82 and 0.86 for SHS and 0.85 for SWLS for age 14 and 16, respectively, and the correlations between the three wellbeing indicators were moderate to high ($r = 0.39–0.75$), supporting their conceptual relatedness and justifying their combination into a single latent construct.

Given that relatively few studies have examined wellbeing specifically in adolescence, and that the conceptualization, measurement, and determinants of wellbeing may vary across age groups, the use of a latent measure supports a more integrated and potentially generalizable approach. This decision is further supported by recent empirical evidence suggesting that broader wellbeing constructs can be meaningfully represented as unidimensional or bifactor models among adolescents[58,59]. Adolescent wellbeing at age 14 was used, and substituted with age 16 in case data was missing at age 14 ($n = 3638$). In contrast to the preregistration, adolescent wellbeing was defined as wellbeing at age 14–16, instead of analyzed individually to increase sample size and hereby power. Supplementary Table S1 shows the sample sizes for the individual ages and the combined ages for all outcome measures.

Where necessary, inversion of item scores was performed to ensure a higher score represents higher wellbeing. Participants with at least two out of three wellbeing measures (life satisfaction, subjective happiness, quality of life) were included. The three measures were standardized prior to analysis. Adolescent wellbeing was then modelled as a latent construct using confirmatory factor analysis (CFA) with these three indicators. A schematic path diagram of the measurement model is provided in Supplementary Fig. S1. Factor scores for each participant were computed using the regression method, which ensures that the covariance matrix of the estimated scores matches the model-based latent covariance structure[60,61]. Model fit indices indicated acceptable to good fit at age 14 and 16: $\chi^2(df) = 422(3)$, $p < .001$, CFI = 0.97, TLI = 0.96, RMSEA = 0.14, McDonald's $\omega = 0.85$, and $\chi^2(df) = 181(2)$, $p < .001$, CFI = 0.98, TLI = 0.98, RMSEA = 0.11, McDonald's $\omega = 0.86$, respectively.

## Outcome measures
Every two to three years, NTR participants 18 years and older receive questionnaires to measure various sociodemographic, health, and lifestyle-related factors, including BMI, physical activity, smoking habits, and wellbeing[50,51]. Since questionnaires are filled out every two to three years, it was possible to select outcome variables based on the two age categories (20–25 and 25–35). Given that data collection for the NTR started in 1986, age categories (20–25 and 25–35 years) were selected based on data availability. In case multiple questionnaires were available within one age category, the questionnaire with the most time between the exposure (i.e. wellbeing at age 14–16) and outcome was chosen. This was done for each outcome individually, meaning that it is possible that for one outcome, e.g. personality, for age category 20–25, a different questionnaire was selected than for a different outcome, e.g. coffee use. The rationale behind this is that 1) not all outcomes are surveyed in all questionnaires, and 2) missing values might occur in individual questionnaires. As a result, the exact number of participants for each outcome differs (see Tables 1–4 and Supplementary Table S1). Various sociodemographic, health, and lifestyle-related factors are included as outcome measures, see Tables 1–4. Due to differences in the measures included across survey waves, sample sizes varied considerably across outcomes (see Tables 1–4). This means that most participants did not contribute data to every analysis.

## Wellbeing outcomes
Identical to the assessment of adolescence wellbeing, wellbeing at the age categories 20–25 and 25–35 was constructed as a single score using factor modelling[55]. Among adults, SWLS and CL were part of all surveys, while SHS was missing in one of the used surveys. In our sample, reliabilities (Cronbach's alpha) ranged from 0.85–0.86 and 0.82–0.89 for SWLS and SHS, respectively. Latent factor scores for that specific survey were created with one less wellbeing measure. However, correlations of the two-measure factors with the three-measure factors were not significantly different than the correlations of the three-measure factors from different surveys (see Supplementary

Table S2), therefore, the latent factor score from any survey can be used with no risk of introducing bias. Additionally, model fit indices for the three-measure factors indicated good fit: $\chi^2(df) = 2.04-15.21(1-2)$, $p < .001$, CFI = 0.99–1.00, TLI = 0.99–1.00, RMSEA = 0.015–0.046, McDonald's $\omega = 0.89-0.90$. The Cronbach $\alpha$ for the two-measure factor was 0.82, showing strong internal consistency.

Flourishing was measured using the Flourishing Scale[62]. The scale contains of eight items questioning an individual's perceived success in multiple life domains. Each item is scored on a scale from 'strong disagreement' (i.e. score 1) to 'strong agreement' (i.e. score 7). The sum score of the eight questions is summarized into one score for flourishing, where higher scores represent higher positive flourishing. Our sample showed good internal consistency using reliability analyses (Cronbach's alpha: 0.82).

## Social outcomes
Personality was assessed using the 60-item NEO-FFI questionnaire, which summarizes the Big Five personality traits using every 12 items of the NEO-FFI questionnaire[63]. The Big Five personality traits consist of neuroticism, extraversion, openness, agreeableness, and conscientiousness. Each individual receives a score for each personality trait, where higher scores represent higher expressions of the respective trait. Reliabilities (Cronbach's alpha) were 0.87, 0.80, 0.74, 0.71, and 0.81 for neuroticism, extraversion, openness, agreeableness, and conscientiousness, respectively.

## Demographic outcomes
Demographic factors consisted of the following: 1) the number of children the individual had at time of the questionnaire, 2) relationship status, which was defined as whether or not the participant indicated to have a sustainable relationship, 3) employment, for which paid work and volunteering were combined, and 4) education, which was defined as having a diploma or not.

## Health outcomes
Health status was defined as whether or not a participant indicated to currently have one or more illnesses. BMI was calculated from self-reported height and weight. Self-rated health was reported by participants with the single item 'In general, how would you rate your health?', which is scored on a five point scale with 1 = 'bad', 2 = 'mediocre', 3 = 'reasonable', 4 = 'good', and 5 = 'excellent'.

## Lifestyle outcomes
Alcohol use was determined by self-reported consumption within the following categories: 'less than 1 glass a day', '1–2 glasses a day', '3–5 glasses a day', '6–10 glasses a day', '11–20 glasses a day', '21–40 glasses a day', and 'more than 40 glasses a day', of which the latter two categories were combined due to low sample sizes. Coffee use was self-reported as cups of coffee a day.

Smoking behaviour was split into smoking cigarettes, smoking e-cigarettes, or whether participants use hash and/or marijuana. Smoking cigarettes was characterized into never smoker, past smoker, and current smoker. E-cigarettes and hash/marijuana were questioned as ever used or not. Drug use was defined as whether or not participants ever experimented with XTC, amphetamines, MDMA or cocaine.

Physical activity is classified into sedentary, moderate, and vigorous exercisers. Total weekly Metabolic Equivalents of Task (MET) minutes for all exercise activities were calculated. A MET score represents the energy that is expended to perform a specific activity relative to the standard resting metabolic rate, equivalating one MET[64,65]. Individuals who did not participate in any exercise activities have a weekly MET hours score of zero. For the remaining individuals, the product of the MET score, weekly frequency and duration was summed across all exercise activities to obtain "total weekly MET hours"[64,65]. Due to the skewed distribution of the continuous MET

score, individuals were classified into three groups: the first category consists of sedentary individuals whose total weekly MET score are lower than 5.0. The second category of moderate exercisers consists of individuals whose total weekly MET score ranges between 5.0 and 30.0. The third category consists of vigorous exercisers whose total weekly MET score is 30.0 or higher, according to previous research[66].

Sleep duration was calculated by subtracting bedtime and wake-up time on weekdays and weekend days of the Munich ChronoType Questionnaire (MCTQ), afterwards the overall sleep duration was calculated[67]. Sleep quality was self-reported using the following item 'I have trouble sleeping', which could be rated with 'not at all', 'somewhat or sometimes', or 'very much so or often' of the Adult Self-Report (ASR) of The Achenbach System of Empirically Based Assessment[68].

Three questions were used to label weight behaviour of participants. Dieting was questioned with 'Have you ever gone on a diet to lose weight or to avoid gaining weight?', where options ranged from 'never', 'a couple of times', 'several times', 'often', and 'always on a diet', of which the latter two categories were combined due to low sample sizes. Next, weight behaviour was surveyed using the question 'How afraid are you to gain weight?' with the possible answers 'not afraid', 'somewhat', 'quite afraid', 'very afraid', and 'extremely afraid', of which the latter two categories were combined due to low sample sizes. Finally, eating behaviour was defined by the question 'Do you usually eat until you are full?', which participant could rate as 'I stop eating before I feel full', 'I stop eating when I feel full', or 'I continue eating even when I feel full'.

## Statistical methods

Multiple regression models using Generalized Estimating Equations (GEE) were conducted to assess associations between adolescent wellbeing and sociodemographic, health, and lifestyle-related outcome. A set of analyses were used, namely between-family analyses including all study participants and within-family analyses including only twins and siblings.

GEE models were chosen to account for familial clustering using family ID as clustering variable. For the within-family approach, sibling fixed-effects were introduced to the model to remove unmeasured confounding due to environments shared by sibling and/or shared genetic background of siblings. The within-family regression formula looks as follows: $Y = \delta_{tw} + M_{tw} + zygosity + zygosity*\delta_{tw} + other\ covariates$, in which $\delta_{tw}$: the individual's deviation from $M_{tw}$, and $M_{tw}$: family mean adolescence wellbeing. The within-family design was only applied for the outcome measures that were shown to be statistically significantly associated with adolescent wellbeing in the between-family design.

Our sample included 10.014 and 4.401 individuals from monozygotic (MZ) and dizygotic (DZ) or sibling families, respectively, across 6981 families in total. Although zygosity information was available, we did not stratify analyses by zygosity, as our focus was on within-family comparisons rather than classical twin modelling. This approach adjusts for all shared genetic and environmental factors that are constant within families, isolating the effect of individual-level deviations from the family mean. As such, it provides a stringent test for within-family associations that may be indicative of causal effects, independent of between-family confounders. Classical twin models (e.g., ACE modelling) would be required to specify the estimated variance attributed to additive genetic and shared and non-shared environmental factors.

Between and within-family analyses were adjusted for age squared at time of outcome measure, age window (i.e. the difference in age between the exposure and outcome measure), and gender in Model 1. As a deviation from the preregistration, Model 1 was additionally adjusted for the outcome measure also surveyed at adolescence age, i.e. the change in outcome measure over time (Model 2). However, outcome measures at adolescence age were not available for the outcome measures 'flourishing', 'children', 'stable relationship', 'drug use', 'smoking e-cigarettes', and 'eating behaviour'. Therefore, no Model 2

could be created for the latter outcome measures. In addition, due to a combination of a limited sample size, low variance in some variables, and the inclusion of multiple predictors, some analyses resulted in computational singularity and as a consequence, no results could be presented.

As an addition to the preregistered analyses, incremental $R^2$ was calculated to assess the proportion of variance in adolescent wellbeing explained by each outcome variable for Model 1 and 2. Incremental $R^2$ was defined as the difference between two nested models: (1) a baseline model including only covariates, and (2) a full model including covariates plus the predictor of interest. For example, the incremental $R^2$ for neuroticism in the between-family analysis was obtained by subtracting the $R^2$ of the baseline model (covariates only) from the $R^2$ of the full model predicting wellbeing from neuroticism and covariates (see Supplementary Table S3–6). For linear and binary outcome measures, the pseudo $R^2$ was calculated, while for multinomial outcomes measures, the Nagelkerke $R^2$ was used, according to previous recommendations for GEE analyses[69,70].

Coefficient-level inference in GEE models was based on Wald tests using standard errors; two-sided $p$ values were obtained from the standard normal distribution. Wald z statistics were computed as the ratio of the estimated coefficient to its standard error. Correction for multiple testing was applied to all analyses using two approaches. First, in alignment with the preregistration, Bonferroni correction. Between-family and within-family analyses were conducted for 50 and 22 outcome measures, respectively, therefore, $p$ values of <0.001 and <0.002, respectively, were considered statistically significant. In addition, FDR-corrected $p$ values are presented since Bonferroni correction has been shown to be too stringent in human sciences[71]. In the current study, statistically significant associations will be presented based on the FDR-corrected $p$ values. After correction for multiple testing, using FDR according to the Benjamin-Hochberg procedure[72], a $p$ value (pFDR) < 0.05 was considered statistically significant.

Mediation analyses were conducted to examine whether associations between adolescent wellbeing and adult outcomes operated through potential mediators measured at age 20–25 using the GEEmediate[73], which accounts for family clustering. Analyses were conducted for the associations that were shown to be significant in the within-family analyses to test how an exposure and outcome are related through potential direct and indirect effects of a mediator variable[74,75]. None of the candidate mediators showed clear evidence of mediation, as indirect effects had confidence intervals including zero across all models. Full details and results are reported in the Supplementary Materials and Supplementary Table S7.

All analyses were performed in R version 4.4.0[76] using the packages *foreign*[77], *lavaan*[78], *dplyr*[79], *gee*[80], *multgee*[81], *GEEmediate*[73], and *forestploter*[82].

No statistical method was used to predetermine sample size. Sample sizes were determined by data availability within the Netherlands Twin Register. No data were excluded from the analyses other than participants with missing exposure or outcome data for a given analysis. The study was observational and therefore not randomized. Investigators were not blinded to exposure or outcome status.

## Reporting summary

Further information on research design is available in the Nature Portfolio Reporting Summary linked to this article.

## Data availability

Data of the participants of the Netherlands Twin Register cannot be made publicly available due to the EU's General Data Protection Regulation, but they are available for researchers via the Netherlands Twin Register data access procedure (https://ntr-data-request.psy.vu.nl/). Detailed information on included data is available on the OSF repository, see https://doi.org/10.17605/OSF.IO/7FYX2.

## Code availability

R scripts for the regression models are available at https://doi.org/10.17605/OSF.IO/7FYX2.

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

## Acknowledgements

The authors would like to thank all participants of the Netherlands Twin Register, who provided data for this study. This work was supported by the NWO Talent Programme – Vici scheme nr. VI.C.211.054 'The Power of Wellbeing' (PI Prof. Dr Bartels), Genetic and Family influences on Adolescent psychopathology and Wellness (NWO 463-06-001), A twin-sib study of adolescent wellness (NWO-VENI 451-04-034), Determinants Of Adolescent Exercise Behavior (NIH-1R01DK092127-01), the NWO large investment grant (NTR: 480-15-001/674), the ZonMW Addiction program (31160008), the ERC Starting 284167, and the ERC consolidator grant (WELLBEING 771057, Prof. Dr Bartels).

## Author contributions

A.J.M.R.G. designed the study, with input from M.B. A.J.M.R.G. analyzed the data, designed the figures and tables, and drafted the paper. All authors contributed to and approved the final version of the paper.

## Competing interests

The authors declare no competing interests.
