## [Transparent Peer Review file · Nature Communications]

Adolescent wellbeing is associated with positive outcomes in early adulthood in a sibling comparison study

Corresponding Author: Dr Anne Geijsen

Version 0:

Reviewer comments:

Reviewer #1

(Remarks to the Author)

This is a potentially important study, using a novel approach, a unique dataset, and with valuable findings. Generally, I find the paper well-written and representing an important contribution to our knowledge about wellbeing and various life outcomes.

However I believe the paper can be improved by addressing the following:

1) The within-family design.

First, it would be helpful to report the number of twin pairs (for each zygosity) and number of sibling pairs in the analyses. Second, the paper would benefit from providing more information about the within-family analyses. From the equation on line 227 it seems the analyses account for the differing genetic similarities for MZ-twins and DZ-twins/siblings, and thus is able to adjust for all genetic variance (unlike a basic sibling-control study, without MZ-twins) and family environment. If so, it would be nice to help readers understand what these analyses do, what is accounted for, and what is left. Also, to what extent can there be causal effects (e.g. due to family processes) that resides in the between-family level, and are excluded from the within-family structures? Is there any way to disentangle the genetic and shared environmental effects in the analyses?

2) Causality and prediction

My general reading of the findings is that wellbeing during adolescence is predictive of several outcomes in adulthood, but the evidence for causality is limited (i.e. few effects when controlling for initial scores and using within-family analyses). The paper might be strengthened by a more elaborate discussion of these two sets of findings. Being able to predict life outcomes 10-15 years later is valuable in itself, and would enable identification of risk groups. It would be nice to see some more discussion of prediction vs causality in the context of the current findings.

3) Minor issues

- The paper says: "For instance, if a trait is more similar within a family compared to unrelated individuals, it suggests a strong genetic component" (line 70). The term 'strong' appears unwarranted, and as stated elsewhere, within-family similarity could also be due to shared environmental factors. Please reformulate.
- The study focuses on age 14-16 (for adolescence), but the Methods section refers to age 14, 16 and 18, and also reports a mean age of 15.9. Please clarify.
- Data from age 14 or 16 were used when only one time point was available, and combined into a mean score when both time points were available. This makes sense in order to maximize sample size and power. However, when mean scores are calculated from two correlated variables, the resulting variable will have reduced variance compared to the two input variables (e.g. participants scoring high at age 14 will tend to score lower at age 16, due to regression to the mean, yielding somewhat lower variance). When parts of the sample have only one measure, and another part have the mean of two time points, and these are analyzed as one total sample, there is a risk that artificial results appear. The paper would benefit from some checking of this issue.
- There are some repetitions, e.g. lines 127-128 appear identical to line 264

Reviewer #2

(Remarks to the Author)

Thank you for the opportunity to review this paper.

This study explored the links between teenage wellbeing (ages 14–16) and sociodemographic, health, and lifestyle outcomes in early adulthood (ages 20–25 and 25–35) using between-family and within-family designs with 14,518 twins and siblings from the Netherlands Twin Register. The authors found positive associations between teenage wellbeing and early adulthood wellbeing, flourishing, conscientiousness, self-rated health, and sleep quality, along with negative associations with neuroticism and smoking. These findings were consistent in sibling fixed-effect analyses, emphasizing the lasting impact of teenage wellbeing on life outcomes, independent of genetic and environmental factors.

This study makes an excellent contribution to the field. The analysis is robust. The paper is well-written and the results are easy to follow. Below, I provide some comments that I hope help strengthen the manuscript:

1. Introduction.

a. I suggest the authors replace the term teenage with adolescence, which is a more academic term, at least in the main text and abstract. Moreover, I encourage them to acknowledge the different stages of adolescence, using terms such as early adolescence and middle adolescence.

b. The authors should define the key terms (teenage/adolescence and wellbeing) at some point in the first paragraphs. Adolescence is increasingly being considered to cover ages 10-24, and what the authors define as teenage (ages 14-16) is middle adolescence (Sawyer et al., 2018). In this study, I think it is more practical and perfectly fine to refer to the age group 20-25 as early adulthood (rather than late adolescence), but I'd use the term middle-adolescence to refer to those aged 14-16. The latter is helpful for later discussions on data gaps, influences of positive outcomes in adulthood preceding age 14-16, etc. (more on this below). As to wellbeing, I'd encourage the authors to briefly mention the main definitions of wellbeing (subjective/hedonic (including cognitive and affective) and psychological/eudaimonic) and clarify that the measures considered are measures of cognitive subjective well-being.

Sawyer, S. M., Azzopardi, P. S., Wickremarathne, D., & Patton, G. C. (2018). The age of adolescence. *The Lancet Child & Adolescent Health*, 2(3), 223–228. [https://doi.org/10.1016/S2352-4642\(18\)30022-1](https://doi.org/10.1016/S2352-4642(18)30022-1)

c. To highlight the importance of the contributions of this study, it may be useful to cite a few other studies that have examined the links between adolescent wellbeing and adult outcomes with less robust methodologies (e.g. Kansky et al., 2016; Goodman et al., 2015), then cite and describe the work by DeNeve and Oswald (2012) as you already do, and then use the last paragraph of the intro to explain more explicitly how this study adds to the existing evidence (e.g. a slightly younger age group compared to the twin study by DeNeve and Oswald (2012), etc.).

Kansky J, Allen JP, Diener E. Early Adolescent Affect Predicts Later Life Outcomes. *Appl Psychol Health Well Being*. 2016 Jul;8(2):192-212. doi: 10.1111/aphw.12068. Epub 2016 Apr 14. PMID: 27075545; PMCID: PMC4931979.

Goodman, A., Joshi, H., Nasim, B., & Tyler, C. (2015, March). Social and emotional skills in childhood and their long-term effects on adult life.

2. Method.

a. Minor consideration. The authors may want to consider using terms for their wellbeing measures that align more closely with the wellbeing definitions briefly described in the intro. Regarding life satisfaction, researchers often distinguish between overall life satisfaction and domain life satisfaction, especially in the field of child and adolescent subjective well-being (Savahl et al., 2021), so the term overall life satisfaction (OLS) may be more indicative of what is being measured (i.e. overall –rather than domain- life satisfaction). For similar reasons, the term overall life evaluation (OLE) –rather than quality of life - may be more adequate to describe what the Cantril Ladder measures.

Savahl, S., Casas, F., & Adams, S. (2021). The structure of children's subjective well-being. *Frontiers in Psychology*. <https://doi.org/10.3389/fpsyg.2021.650691>

b. The authors should report the Cronbach Alpha of the scales.

c. To further support the decision to use a latent wellbeing measure derived from the three wellbeing measures, the authors can cite recent evidence from the field of child/adolescent wellbeing providing support for the consideration of conceptually broader constructs (wellbeing and subjective well-being) as unidimensional constructs (Savahl et al., 2023; Khanna et al. 2024)

Savahl, S., Casas, F. & Adams, S. Considering a Bifactor Model of Children's Subjective Well-Being Using a Multinational Sample. *Child Ind Res* 16, 2253–2278 (2023). <https://doi.org/10.1007/s12187-023-10058-6>

Khanna, D., Black, L., Panayiotou, M. et al. Conceptualising and Measuring Adolescents' Hedonic and Eudemonic Wellbeing: Discriminant Validity and Dimensionality Concerns. *Child Ind Res* 17, 551–579 (2024).

<https://doi.org/10.1007/s12187-024-10106-9>

3. Results.

a. This looks good.

4. Discussion

a. At some point the authors discuss the potential role of prior influences preceding this age group (Age 14-16). This could still take place during teenage/adolescence. That is why I think it'd help to use the term middle adolescence to refer to ages 14-16 and acknowledge that wellbeing and its correlates vary across different childhood and adolescence stages (e.g. see Marquez et al., 2024).

b. The call for collecting more and better longitudinal data from birth may emphasize that this is especially needed in those socio-cultural context where data is far more scarce (i.e. non-high-income non-Western countries, where results may be different; Marquez et al., 2024)

Marquez, J., Taylor, L., Boyle, L., Zhou, W., & De Neve, J.E. (2024). Child and Adolescent Well-Being: Global Trends, Challenges and Opportunities. *World Happiness Report, 2024*. <https://doi.org/10.18724/whr-91b0-ek06>

(Remarks to the Author)

The present study examines the associations between adolescent wellbeing at ages 14-16 and early adult demographic, social, health, and lifestyle outcomes, accounting for the potential influence of shared genetic and environmental influences in analyses. The authors report significant associations between teenage well-being and adult outcomes, including well-being, flourishing, conscientiousness, self-rated health, sleep quality, neuroticism, and smoking. This preregistered study is strengthened by the use of a large, longitudinal dataset with a broad range of outcome measures for twins and their siblings. The findings are noteworthy, given that they emphasise the potential importance of adolescent wellbeing for later-life outcomes across multiple domains of functioning – an under-researched area, as discussed by the authors.

However, we have identified several issues relating to the conceptualisation of key terminology, measurement of wellbeing, reporting of statistical information, and interpretation of study findings that require further consideration.

Key points for consideration:

- The study leverages a causal inference approach (a sibling comparison approach) to understand the effect of teenage wellbeing on later outcomes. This design is a major strength of the study. It is, therefore, worth highlighting this approach in the title and abstract. Providing a clear rationale for this approach in the introduction and examining its strengths and limitations in the discussion would also strengthen this piece.
- Relatedly, it would be useful to compare the within- and between-family findings more consistently and clearly. The differences between the two could be informative of potential causal effects (or, conversely, likely environmental and genetic confounding). It would be worth adding a section to the results that compares these effects directly. The discussion should then pick up on these more clearly, too.
- The hypotheses should be stated clearly in the introduction and referred back to in the discussion.
- We note that in the preregistration for this study, the authors outline planned mediation analyses using a panel of potential mediating variables measured at 20-25 years of age. It would be helpful if the authors provided clear information in the manuscript on whether these preregistered mediation analyses were conducted.
- Relatedly, the manuscript provides a link to access reproducible code; however, no scripts are currently visible. Is it possible for the authors to amend this?

Abstract

- Please add statistics for key findings.

Introduction

- The authors do not provide clear definitions for adolescence or wellbeing in the present investigation. Additionally, the terms teenage and adolescence are used interchangeably throughout the manuscript. It would be helpful to include 1) a definition of the developmental phase investigated in this study, as well as how this definition applies to the specific age range covered here, i.e., 14-16 years, and 2) a definition of adolescent wellbeing, information on how it has been conceptualised in prior research, and how it is assessed in the present investigation.
- The authors discuss only one study in detail examining the impact of adolescent wellbeing on adult income (De Neve & Oswald, 2012). While it can be assumed that the examination of a broad range of outcome measures in this study is due to a lack of prior work investigating the impact of teenage wellbeing on other later-life outcomes, explicitly stating this – or highlighting additional investigations on the associations between teenage wellbeing and adult demographic, social, health, and lifestyle outcomes – would be beneficial.

Methods

- More detailed demographics, including SES and ethnicity, would be helpful. How representative is the sample? Detailed information on missingness in Table 1 would also be helpful.
- We noted several issues pertaining to the measurement of wellbeing in this investigation:
 - 1) As it stands, it is not clear why the measures life satisfaction, subjective happiness, and quality of life, were chosen to reflect wellbeing. We recommend that the authors justify this choice by a) providing information on the reliability and validity of the measures chosen in the target population and for the measurement purpose; b) by including additional statistical information on the relationships between these measures (e.g., reporting correlations between these measures).
 - 2) Insufficient details are provided regarding the construction of the wellbeing latent score (page 7). We recommend that the authors report the results of the factor modelling to construct the single well-being score, including comprehensive model fit statistics.
 - 3) The authors state that the Netherlands Twin Register contains wellbeing measures at ages 14, 16, and 18 (page 6). However, the present investigation only includes wellbeing assessed at ages 14 -16. Why was this the case?
- It would be helpful to provide additional information regarding the choice of specific age categories for adult outcomes (i.e., 20-25 and 25-35).

Results

- The results section would benefit from greater explanation, e.g. not just stating associations but interpreting them for the reader, too.
- The authors state on page 12 that they calculated the incremental R² as an addition to preregistered analyses to examine the “difference between the R² of the regressions with and without the outcome measure”. However, it is unclear as to which models are compared based on R² statistics to determine this value. We recommend that further details be provided on this.

Discussion

- The discussion would benefit from a clearer paragraph structure throughout that crystallises key messages for the reader. E.g., a strengths and limitations paragraph would also be useful. The writing could be less descriptive and provide more evaluation and analysis. For instance, in the first paragraph, rather than re-stating technical findings of “sibling fixed effects”, highlighting key findings in relationship to hypotheses as well as implications for theory and practice would be useful.
- In the opening paragraph of the discussion, the authors discuss the significant between-family associations found in this study. However, it could be clearer whether the associations described here are specifically those found after controlling for these outcome variables assessed at teenage. This also applies to the description of results in the rest of the discussion section.
- Despite evidence for differences in findings between the 20-25 and 25-35 age brackets (e.g., teenage wellbeing only predicts conscientiousness at age 20-25, but not at age 25-35), differences between age groups are not explicitly outlined or discussed in depth in the discussion. It would be helpful to discuss these differences in further detail. Relatedly, there are some inconsistencies in reporting regarding the association between teenage wellbeing and adult hash and marijuana use (this was significant at ages 20-25 and non-significant at ages 25-35).
- On page 20 of the discussion, the authors note that the associations between teenage wellbeing and adult outcomes “underscores the potential for intervention protocols aimed at improving wellbeing at any stage of life.” Given that the present results support an association between teenage wellbeing and adult outcomes, we suggest that the authors’ interpretation is amended to reflect the results better.
- Conscientiousness is incorrectly referred to as consciousness at several points throughout the manuscript, including in the abstract and discussion.

Reviewer #4

(Remarks to the Author)

Version 1:

Reviewer comments:

Reviewer #1

(Remarks to the Author)

I believe the authors have addressed all my concerns, and that the paper now will become a very nice contribution to the literature.

Reviewer #2

(Remarks to the Author)

Thank you. My concerns have been addressed in the revisions and I do not have anything else to add.

Reviewer #3

(Remarks to the Author)

Thank you for the opportunity to review the revised manuscript, as well as the authors’ response letter. We appreciate the authors’ efforts to address the points raised in our initial round of review.

The authors now clearly outline the differences between a between-family and within-family design, as well as discuss the strengths and weaknesses of the within-family approach. The manuscript also provides definitions of wellbeing and adolescence. We also note that the authors have made their analysis code publicly available and that the abstract now includes key statistics.

There are several remaining points that we believe would benefit from additional clarification and/or revision:

- We appreciate that the authors now state their hypotheses in the introduction. However, given the wide range of outcomes examined in the present study, it would be helpful if the authors could 1) explicitly comment on why such a broad range of outcome measures were chosen and 2) indicate whether any specific hypotheses were made for specific outcomes. If no such hypotheses were made due to a lack of prior literature, this could be stated explicitly to aid readers’ interpretation.
- We understand that the authors chose not to report the results of the mediation analyses to maintain the readability of the manuscript. However, given that these analyses were preregistered, omitting these analyses is perhaps not optimal from a standpoint of complete reporting and transparency. We suggest that this deviation from analysis plans is explicitly outlined as a deviation from the preregistered analyses in the Methods section.
- We appreciate that there were varying sample sizes per outcome due to differences in the measures included in each survey, and that most participants were not surveyed for every item. However, looking at Table 1, sample sizes seem to vary considerably across outcome variables, and some appear quite small (e.g., sleep duration at age 25-35: N = 29). We

suggest that the authors comment on this variation and consider whether analyses for certain outcomes may have been underpowered.

- When the authors state that the regression method was used to create the factor score, do they mean that a confirmatory factor analysis (CFA) was first conducted to model the latent construct, followed by using regression to compute individual scores? Clarifying this process and including a path diagram would help readers better understand the measurement approach.
- We thank the authors for providing additional information regarding the incremental R2. However, it remains unclear how an R2 can be calculated "without the outcome variable".
- We appreciate that the authors have now commented on the differing findings between adult age groups. However, it would be helpful if the authors could elaborate further on what is meant by "potential developmental shifts or changing environmental influences across early adulthood" and how this might help to interpret the observed results.

Reviewer #4

(Remarks to the Author)

Version 2:

Reviewer comments:

Reviewer #3

(Remarks to the Author)

We thank the authors for their thoughtful response to our feedback. Our concerns have been addressed. This is a minor point, but we recommend aligning the CFA path diagram with field conventions, e.g., by showing factor loadings and variances.

Reviewer #4

(Remarks to the Author)

NCOMMS-24-78782 - Adolescent wellbeing as a catalyst for positive outcomes in early adulthood using a sibling sample

Responses to Reviewers

We are grateful to the reviewers for their compliments, the thorough evaluation of our manuscript and providing valuable feedback. We have implemented the comments and issues raised accordingly. In the revised manuscript, the changes made are noted in red font colour. We hope you will find the revised manuscript suitable for publication in your esteemed journal.

Reviewer #1:

1) The within-family design.

First, it would be helpful to report the number of twin pairs (for each zygosity) and number of sibling pairs in the analyses.

Second, the paper would benefit from providing more information about the within-family analyses. From the equation on line 227 it seems the analyses account for the differing genetic similarities for MZ-twins and DZ-twins/siblings, and thus is able to adjust for all genetic variance (unlike a basic sibling-control study, without MZ-twins) and family environment. If so, it would be nice to help readers understand what these analyses do, what is accounted for, and what is left. Also, to what extent can there be causal effects (e.g. due to family processes) that resides in the between-family level, and are excluded from the within-family structures? Is there any way to disentangle the genetic and shared environmental effects in the analyses?

We thank the reviewer for this suggestion. While we do include both MZ and DZ twins (as well as non-twin siblings) in our within-family design, our primary goal was to adjust for shared familial influences using family-level clustering, not to decompose variance by zygosity. Therefore, we now report the total number of twin and sibling pairs to improve transparency, but we avoid breaking this down by zygosity in the main text to prevent confusion, as these distinctions were not used in the core analyses. We have added a clarifying sentence to the Methods section (see line 286-295) and now discuss in the Discussion (see line 605-610) that classical twin modeling (e.g., ACE models) would be a logical next step for future studies.

“Our sample included 10.014 and 4.401 individuals from monozygotic (MZ) and dizygotic (DZ) or sibling families, respectively, across 6981 families in total. Although zygosity information was available, we did not stratify analyses by zygosity, as our focus was on within-family comparisons rather than classical twin modeling. This approach adjusts for all shared genetic and environmental factors that are constant within

families, isolating the effect of individual-level deviations from the family mean. As such, it provides a stringent test for within-family associations that may be indicative of causal effects, independent of between-family confounders. Classical twin models (e.g., ACE modelling) would be required to specify the estimated variance attributed to additive genetic and shared and non-shared environmental factors.”

and

“Although our within-family design adjusts for shared familial influences, including both genetic and shared environmental factors, it does not allow us to decompose these influences separately. Future studies could apply classical twin models (e.g., ACE modeling) to estimate the proportion of variance attributable to additive genetic, shared environmental, and non-shared environmental factors, particularly for those associations that remained robust in the within-family analyses.”

2) Causality and prediction

My general reading of the findings is that wellbeing during adolescence is predictive of several outcomes in adulthood, but the evidence for causality is limited (i.e. few effects when controlling for initial scores and using within-family analyses).

The paper might be strengthened by a more elaborate discussion of these two sets of findings. Being able to predict life outcomes 10-15 years later is valuable in itself, and would enable identification of risk groups. It would be nice to see some more discussion of prediction vs causality in the context of the current findings.

We have added a paragraph discussing the necessary distinction between prediction and causality, see lines 551-562:

“Furthermore, the distinction between prediction and causality is critical. Our findings show that adolescent wellbeing is consistently associated with multiple adult outcomes, with associations remaining robust over a follow-up period of 10–15 years, underscoring its predictive utility. When controlling for baseline levels and shared familial factors via within-family analyses, the number of significant associations was reduced, suggesting that some associations may reflect confounding by stable family-level influences or pre-existing differences. However, several associations, most notably with wellbeing, flourishing, and sleep quality at age 20–25, remained statistically significant even after these stringent adjustments. These findings offer stronger evidence for potential direct effects of adolescent wellbeing and highlight its value for early identification and preventive interventions. Future work integrating predictive and causal modelling could help clarify when and how wellbeing exerts long-term influence on life trajectories.”

3) Minor issues

- The paper says: “For instance, if a trait is more similar within a family compared to

unrelated individuals, it suggests a strong genetic component” (line 70). The term ‘strong’ appears unwarranted, and as stated elsewhere, within-family similarity could also be due to shared environmental factors. Please reformulate.

Thank you for the highlighting this overstatement, we changed it to a more nuanced explanation, see lines 89-90:

“For instance, if a trait is more similar within a family compared to unrelated individuals, it suggests the influence of familial factors, such as shared genes and/or shared environment.”

- The study focuses on age 14-16 (for adolescence), but the Methods section refers to age 14, 16 and 18, and also reports a mean age of 15.9. Please clarify.

We initially intended to examine data collected at ages 14, 16, and 18 separately. However, due to limited data availability at some of these time points, we decided to focus on the adolescent data collected between ages 14 and 16. Questionnaires were distributed in waves to adolescent twins and their siblings, following parental consent. Because of the timing of consent procedures and data collection, the average age of respondents was slightly higher than the target range, with a mean of 15.9 years.

We have now added this to the Methods section for clarity, see lines 118-126:

“We used adolescent data collected by the NTR around ages 14 to 16, with a focus on participants with available wellbeing and outcome measures. The sample size to assess teenage wellbeing, using three wellbeing measures, resulted in 14,518 adolescents with available teenage wellbeing measures. Characteristics of the study population are described in Table 1. About 58% of the study population are women. The mean (SD) age of the adolescents was 15.9 (1.8) years. Questionnaires were distributed in waves to adolescent twins and their siblings, following parental consent. Because of the timing of consent procedures and data collection, the average age of respondents was slightly higher than the target range, with a mean of 15.9 years.”

- Data from age 14 or 16 were used when only one time point was available, and combined into a mean score when both time points were available. This makes sense in order to maximize sample size and power. However, when mean scores are calculated from two correlated variables, the resulting variable will have reduced variance compared to the two input variables (e.g. participants scoring high at age 14 will tend to score lower at age 16, due to regression to the mean, yielding somewhat lower variance). When parts of the sample have only one measure, and another part have the mean of two time points, and these are analyzed as one total sample, there is a risk that artificial results appear. The paper would benefit from some checking of this issue.

We thank the reviewer for raising this important concern. To address the potential bias introduced by combining wellbeing scores across age 14 and 16, we first examined the

distribution of adolescent wellbeing separately for participants with data at age 14 only, age 16 only, or both (mean score). As shown in Figure 1, the distributions were highly similar in median and spread across groups. The correlation between age 14 and 16 scores was moderate ($r = 0.51$), supporting their conceptual and statistical alignment.

Figure 1 Distribution of adolescent wellbeing by age selected

Nevertheless, we agree that the original strategy could have introduced heterogeneity. We therefore re-ran all analyses using a revised approach: wellbeing scores from age 14 were prioritized ($n=10,880$), and scores from age 16 were only used when age 14 data were missing ($n=3,638$). All tables, figures, and corresponding text have been updated accordingly. Although some minor changes in statistical significance were observed (e.g., BMI became significant in one model; sleep quality at age 25–35 in within-family analyses was no longer significant), the overall pattern of results and study conclusions remained unchanged.

- There are some repetitions, e.g. lines 127-128 appear identical to line 264

Thank you for this observation, we have removed the information in the original lines 127-128.

Reviewer #2:

Thank you for the opportunity to review this paper.

This study explored the links between teenage wellbeing (ages 14–16) and sociodemographic, health, and lifestyle outcomes in early adulthood (ages 20–25 and 25–35) using between-family and within-family designs with 14,518 twins and siblings from the Netherlands Twin Register. The authors found positive associations between teenage wellbeing and early adulthood wellbeing, flourishing, conscientiousness, self-rated health, and sleep quality, along with negative associations with neuroticism and smoking. These findings were consistent in sibling fixed-effect analyses, emphasizing the lasting impact of teenage wellbeing on life outcomes, independent of genetic and environmental factors.

This study makes an excellent contribution to the field. The analysis is robust. The paper is well-written and the results are easy to follow. Below, I provide some comments that I hope help strengthen the manuscript:

1. Introduction.

a. I suggest the authors replace the term teenage with adolescence, which is a more academic term, at least in the main text and abstract. Moreover, I encourage them to acknowledge the different stages of adolescence, using terms such as early adolescence and middle adolescence.

Thank you for the suggestion, we have changed teenage to adolescence throughout the whole manuscript, with acknowledging the difference stages of adolescence, for example see lines 96-100:

“Therefore, the objective of the current study is to investigate the association between middle-adolescent wellbeing (age 14-16) and sociodemographic, health and lifestyle outcomes in early adulthood (age 20-25 and 25-35) using both between-family and within-family designs among 14518 twins and their siblings of the Netherlands Twin Register.”

b. The authors should define the key terms (teenage/adolescence and wellbeing) at some point in the first paragraphs. Adolescence is increasingly being considered to cover ages 10-24, and what the authors define as teenage (ages 14-16) is middle adolescence (Sawyer et al., 2018). In this study, I think it is more practical and perfectly fine to refer to the age group 20-25 as early adulthood (rather than late adolescence), but I'd use the term middle-adolescence to refer to those aged 14-16. The latter is helpful for later discussions on data gaps, influences of positive outcomes in adulthood preceding age 14-16, etc. (more on this below). As to wellbeing, I'd encourage the authors to briefly mention the main definitions of wellbeing (subjective/hedonic (including cognitive and affective) and psychological/eudaimonic) and clarify that the measures considered are measures of cognitive subjective well-being.

Sawyer, S. M., Azzopardi, P. S., Wickremarathne, D., & Patton, G. C. (2018). The age of adolescence. *The Lancet Child & Adolescent Health*, 2(3), 223–228.

[https://doi.org/10.1016/S2352-4642\(18\)30022-1](https://doi.org/10.1016/S2352-4642(18)30022-1)

We thank the reviewer for the suggestions in point a and b, we now provide definitions for our key concept, and address teenage/adolescence is a more precise matter. Please see lines 46-54:

“Wellbeing, often considered an umbrella term, encompasses various aspects of positive and negative life evaluations, emotional states, and a person’s sense of meaning (OECD, 2013). A conceptual distinction can be made between subjective (or hedonic) and psychological (or eudaimonic) wellbeing (Deci & Ryan, 2008; Ryff, 1989), where subjective wellbeing is defined as the cognitive and affective evaluation of a person’s life (Deci & Ryan, 2008; Diener et al., 1985). Psychological wellbeing is summarized as positive functioning and purpose in life (Ryff, 1989). Many of these measures of wellbeing correlate moderately to strongly, suggesting an underlying broad wellbeing factor (Bartels & Boomsma, 2009; Baselmans et al., 2019; Baselmans & Bartels, 2018).

Adolescents constitute over one-sixth of the global population (Unicef, 2024), where adolescence is the period from age 10 up to 24 years of age (Sawyer et al., 2018).”

c. To highlight the importance of the contributions of this study, it may be useful to cite a few other studies that have examined the links between adolescent wellbeing and adult outcomes with less robust methodologies (e.g. Kansky et al., 2016; Goodman et al., 2015), then cite and describe the work by DeNeve and Oswald (2012) as you already do, and then use the last paragraph of the intro to explain more explicitly how this study adds to the existing evidence (e.g. a slightly younger age group compared to the twin study by DeNeve and Oswald (2012), etc.).

Kansky J, Allen JP, Diener E. Early Adolescent Affect Predicts Later Life Outcomes. *Appl Psychol Health Well Being*. 2016 Jul;8(2):192-212. doi: 10.1111/aphw.12068. Epub 2016 Apr 14. PMID: 27075545; PMCID: PMC4931979.

Goodman, A., Joshi, H., Nasim, B., & Tyler, C. (2015, March). Social and emotional skills in childhood and their long-term effects on adult life.

Thank you for highlighting these publications. We have included the results of Kansky et al. (2016) in our introduction, see lines 71-73:

“To illustrate, among 186 adolescents of age 14, positive affect was shown to be positively associated with friendship attachment and career outcomes, and inversely associated with loneliness and anxiety at ages 23-35 (Kansky et al., 2016).”

Although the comprehensive review by Goodman et al. (2015) provides valuable insights, we consider it less directly applicable to the current study. Chapter 2 focuses

on identifying social and emotional skills that predict later life outcomes; however, the majority of the cited studies examine skills assessed during childhood. The subset of studies that align with our target age range (adolescents aged 14 to 16) primarily investigate constructs such as self-esteem and personality as predictors, rather than (components of) wellbeing. While these constructs are related to wellbeing, our focus is specifically on wellbeing itself, and we therefore aim to maintain a clear conceptual distinction and decided not to include the proposed reference.

2. Method.

a. Minor consideration. The authors may want to consider using terms for their wellbeing measures that align more closely with the wellbeing definitions briefly described in the intro. Regarding life satisfaction, researchers often distinguish between overall life satisfaction and domain life satisfaction, especially in the field of child and adolescent subjective well-being (Savahl et al., 2021), so the term overall life satisfaction (OLS) may be more indicative of what is being measured (i.e. overall –rather than domain- life satisfaction). For similar reasons, the term overall life evaluation (OLE) –rather than quality of life - may be more adequate to describe what the Cantril Ladder measures.

Savahl, S., Casas, F., & Adams, S. (2021). The structure of children's subjective well-being. *Frontiers in Psychology*. <https://doi.org/10.3389/fpsyg.2021.650691>

Thank you for the valuable suggestion regarding the terminology and justification of our wellbeing measures. We use the term “wellbeing” as an umbrella construct that integrates three well-established measures: life satisfaction, subjective happiness, and quality of life, consistent with prior research (Bartels et al., 2013; Bartels, 2015).

To acknowledge the reviewer's point about terminology, we clarify that the life satisfaction measure corresponds to overall life satisfaction, and the quality of life measure aligns with overall life evaluation as assessed by the Cantril Ladder (Savahl et al., 2021). We consider these complementary facets together to capture a broad, reliable latent construct of adolescent wellbeing.

Our manuscript now explicitly states this in the Methods section (lines 142-149), alongside factor analytic evidence supporting the validity of combining these measures into a single latent score. This approach follows recent work emphasizing the benefit of integrating multiple indicators to represent adolescent wellbeing comprehensively (Khanna et al., 2024; Savahl et al., 2023).

b. The authors should report the Cronbach Alpha of the scales.

We have now added the Cronbach's alphas for the scales in the used sample in the Methods section.

c. To further support the decision to use a latent wellbeing measure derived from the three wellbeing measures, the authors can cite recent evidence from the field of

child/adolescent wellbeing providing support for the consideration of conceptually broader constructs (wellbeing and subjective well-being) as unidimensional constructs (Savahl et al., 2023; Khanna et al. 2024)

Savahl, S., Casas, F. & Adams, S. Considering a Bifactor Model of Children’s Subjective Well-Being Using a Multinational Sample. *Child Ind Res* 16, 2253–2278 (2023).

<https://doi.org/10.1007/s12187-023-10058-6>

Khanna, D., Black, L., Panayiotou, M. et al. Conceptualising and Measuring Adolescents’ Hedonic and Eudemonic Wellbeing: Discriminant Validity and Dimensionality Concerns. *Child Ind Res* 17, 551–579 (2024).

<https://doi.org/10.1007/s12187-024-10106-9>

Thank you for the suggestion, we have updated the Methods section, see lines 150-156:

“Given that relatively few studies have examined wellbeing specifically in adolescence, and that the conceptualization, measurement, and determinants of wellbeing may vary across age groups, the use of a latent measure supports a more integrated and potentially generalizable approach. This decision is further supported by recent empirical evidence suggesting that broader wellbeing constructs can be meaningfully represented as unidimensional or bifactor models among adolescents (Khanna et al., 2024; Savahl et al., 2023).”

3. Results.

a. This looks good.

4. Discussion

a. At some point the authors discuss the potential role of prior influences preceding this age group (Age 14-16). This could still take place during teenage/adolescence. That is why I think it’d help to use the term middle adolescence to refer to ages 14-16 and acknowledge that wellbeing and its correlates vary across different childhood and adolescence stages (e.g. see Marquez et al., 2024).

Thank you for the suggestion. We have elaborated on the topic in the discussion, see lines 583-594:

“Beyond sample size considerations, notable differences emerged between the 20–25 and 25–35 age groups. For example, adolescent wellbeing predicted conscientiousness and cannabis use in the younger group but not in the older group, suggesting potential developmental shifts or changing environmental influences across early adulthood.

Further, it is important to note that ages 14–16 correspond to middle adolescence, and wellbeing and its correlates may vary across different developmental stages (Marquez et al., 2024) highlighting that some influences on wellbeing at this age may have originated earlier in childhood or adolescence. These findings highlight the importance of further exploring age-specific mechanisms in future research, including life exposures and outcomes before, during adolescence and beyond age 35 to study in future studies to

explore the relationship whether adolescent wellbeing and life outcomes in later adulthood.”

b. The call for collecting more and better longitudinal data from birth may emphasize that this is especially needed in those socio-cultural context where data is far more scarce (i.e. non-high-income non-Western countries, where results may be different; Marquez et al., 2024)

Marquez, J., Taylor, L., Boyle, L., Zhou, W., & De Neve, J.E. (2024). Child and Adolescent Well-Being: Global Trends, Challenges and Opportunities. World Happiness Report, 2024. <https://doi.org/10.18724/whr-91b0-ek06>

Thank you for the suggestion, we agree this should be explicitly be mentioned. Please see lines 580-582:

“Further studies with larger and more diverse populations, especially among non-Western and lower-income populations that are currently underrepresented in literature (Marquez et al., 2024), may help to address these issues and allow for more robust modeling of the relationships of interest.”

Reviewer #3:

The present study examines the associations between adolescent wellbeing at ages 14-16 and early adult demographic, social, health, and lifestyle outcomes, accounting for the potential influence of shared genetic and environmental influences in analyses. The authors report significant associations between teenage well-being and adult outcomes, including well-being, flourishing, conscientiousness, self-rated health, sleep quality, neuroticism, and smoking. This preregistered study is strengthened by the use of a large, longitudinal dataset with a broad range of outcome measures for twins and their siblings. The findings are noteworthy, given that they emphasise the potential importance of adolescent wellbeing for later-life outcomes across multiple domains of functioning – an under-researched area, as discussed by the authors.

However, we have identified several issues relating to the conceptualisation of key terminology, measurement of wellbeing, reporting of statistical information, and interpretation of study findings that require further consideration.

Key points for consideration:

- The study leverages a causal inference approach (a sibling comparison approach) to understand the effect of teenage wellbeing on later outcomes. This design is a major strength of the study. It is, therefore, worth highlighting this approach in the title and abstract. Providing a clear rationale for this approach in the introduction and examining its strengths and limitations in the discussion would also strengthen this piece.
- Relatedly, it would be useful to compare the within- and between-family findings more consistently and clearly. The differences between the two could be informative of potential causal effects (or, conversely, likely environmental and genetic confounding). It would be worth adding a section to the results that compares these effects directly. The discussion should then pick up on these more clearly, too.

We have adjusted the title, abstract, and introduction to make the study design more obvious. We have also added the hypotheses to the introduction and discussion. In general, subheadings in the discussion were added to make it more coherent, clearly stating the strengths and limitations, as well as addressing the overlap of the within- and between-family findings in more detail. More information on the chosen model can be found in line 286-295 (Methods) and the Discussion (see line 605-610):

“Our sample included 10,014 and 4,401 individuals from monozygotic (MZ) and dizygotic (DZ) or sibling families, respectively, across 6,981 families in total. Although zygosity information was available, we did not stratify analyses by zygosity, as our focus was on within-family comparisons rather than classical twin modeling. This approach

adjusts for all shared genetic and environmental factors that are constant within families, isolating the effect of individual-level deviations from the family mean. As such, it provides a stringent test for within-family associations that may be indicative of causal effects, independent of between-family confounders. Classical twin models (e.g., ACE modelling) would be required to specify the estimated variance attributed to additive genetic and shared and non-shared environmental factors.”

and

“Although our within-family design adjusts for shared familial influences, including both genetic and shared environmental factors, it does not allow us to decompose these influences separately. Future studies could apply classical twin models (e.g., ACE modeling) to estimate the proportion of variance attributable to additive genetic, shared environmental, and non-shared environmental factors, particularly for those associations that remained robust in the within-family analyses.”

- The hypotheses should be stated clearly in the introduction and referred back to in the discussion.

We thank the reviewer for this important suggestion. We have now clearly stated our hypotheses in the Introduction section (see lines 100-104) and explicitly referred back to them in the Discussion when interpreting the results (see lines 468-471). This should help guide the reader more clearly through the rationale, findings, and implications of our study.

“Our hypothesis is that higher adolescent wellbeing is positively associated with sociodemographic, health, and lifestyle outcomes in early adulthood. We further hypothesize that some of these associations will remain statistically significant after accounting for shared familial factors through within-family analyses, suggesting potential causal relationships.”

and

“In line with our preregistered expectations, several sociodemographic, health, and lifestyle outcomes were associated with adolescent wellbeing. However, not all outcomes showed significant associations, and the direction of effects varied across predictors.”

- We note that in the preregistration for this study, the authors outline planned mediation analyses using a panel of potential mediating variables measured at 20-25 years of age. It would be helpful if the authors provided clear information in the manuscript on whether these preregistered mediation analyses were conducted.

We thank the reviewer for pointing this out. In the preregistration, we indeed outlined plans for mediation analyses using potential mediating variables measured at ages 20–25. These analyses were not included in the current manuscript to maintain focus and readability, as including them would have significantly expanded the scope and

complexity of the paper. However, we still consider these analyses important and worthwhile, and we plan to report them in a separate follow-up study. We now clarify this in the manuscript (see Discussion, line 563-567).

“As preregistered, we also planned mediation analyses to explore potential pathways through which early-life factors influence adult outcomes via mediators measured at ages 20–25. To ensure the clarity and coherence of the current manuscript, we chose not to include these additional analyses here. However, we consider them an important next step and aim to report them in a separate follow-up study.”

- Relatedly, the manuscript provides a link to access reproducible code; however, no scripts are currently visible. Is it possible for the authors to amend this?

Thank you for pointing this out. We have now uploaded all analysis scripts and data processing code to the repository linked in the manuscript.

Abstract

- Please add statistics for key findings.

Thank you for this helpful suggestion. We fully agree that including effect sizes improves the clarity and informativeness of the abstract. At the same time, given the word limit and the number of significant findings across different models and age groups, we had to strike a careful balance between statistical detail and readability. In the revised abstract, we have now included key effect size ranges (β s) and significance levels (pFDR) for the primary outcomes that remained robust across models. We hope this provides sufficient statistical context while maintaining a clear and accessible summary of the study’s contributions.

Introduction

- The authors do not provide clear definitions for adolescence or wellbeing in the present investigation. Additionally, the terms teenage and adolescence are used interchangeably throughout the manuscript. It would be helpful to include 1) a definition of the developmental phase investigated in this study, as well as how this definition applies to the specific age range covered here, i.e., 14-16 years, and 2) a definition of adolescent wellbeing, information on how it has been conceptualised in prior research, and how it is assessed in the present investigation.

We now provide definitions for our key concept, and address teenage/adolescence is a more precise matter. Please see lines 46-54:

“Wellbeing, often considered an umbrella term, encompasses various aspects of positive and negative life evaluations, emotional states, and a person’s sense of meaning (OECD, 2013). A conceptual distinction can be made between subjective (or hedonic) and psychological (or eudaimonic) wellbeing (Deci & Ryan, 2008; Ryff, 1989), where subjective wellbeing is defined as the cognitive and affective evaluation of a

person's life (Deci & Ryan, 2008; Diener et al., 1985). Psychological wellbeing is summarized as positive functioning and purpose in life (Ryff, 1989). Many of these measures of wellbeing correlate moderately to strongly, suggesting an underlying broad wellbeing factor (Bartels & Boomsma, 2009; Baselmans et al., 2019; Baselmans & Bartels, 2018).

Adolescents constitute over one-sixth of the global population (Unicef, 2024), where adolescence is the period from age 10 up to 24 years of age (Sawyer et al., 2018)."

- The authors discuss only one study in detail examining the impact of adolescent wellbeing on adult income (De Neve & Oswald, 2012). While it can be assumed that the examination of a broad range of outcome measures in this study is due to a lack of prior work investigating the impact of teenage wellbeing on other later-life outcomes, explicitly stating this – or highlighting additional investigations on the associations between teenage wellbeing and adult demographic, social, health, and lifestyle outcomes – would be beneficial.

We have now included the results of Kansky et al. (2016) in our introduction, see lines 71-73, and expanded our summary in the introduction, see lines 94-96:

"To illustrate, among 186 adolescents of age 14, positive affect was shown to be positively associated with friendship attachment and career outcomes, and inversely associated with loneliness and anxiety at ages 23-35 (Kansky et al., 2016)."

and

"Summarizing, there is a notable gap in understanding the role of adolescent wellbeing in shaping later life trajectories as earlier studies are limited in sample size and the investigated outcome measures and age windows."

Methods

- More detailed demographics, including SES and ethnicity, would be helpful. How representative is the sample? Detailed information on missingness in Table 1 would also be helpful.

We appreciate the reviewer's suggestions. Unfortunately, detailed SES and ethnicity information is not available for the current cohort. However, several demographic indicators (e.g., BMI, education, and marital status) are included as outcome variables in our analyses.

We have added a note in Table 1 to clarify that not all participants have data on all outcomes, due to the structure of follow-up assessments. For each outcome, we selected a follow-up survey that allowed for the longest time window between adolescent wellbeing and the respective outcome measure. As a result, different

subsamples are included across outcomes. Not all participants were invited to all follow-up surveys, and not all outcome measures were included in every survey. As a result, the number of individuals included per outcome varies, and most participants were not surveyed for every outcome. We, therefore, chose not report exact sample sizes per outcome in this table, as doing so may give a misleading impression of low response rates.

To address representativeness, we now include a reference in the Methods section to earlier research showing that the NTR is broadly representative of the Dutch population, though women and highly educated individuals are somewhat overrepresented, and non-Caucasian individuals are underrepresented, see lines 113-116:

“NTR may be seen as a good representation of the majority of the Dutch population, although females and higher educated individuals are somewhat overrepresented and non-Caucasian individuals are underrepresented (Willemsen et al., 2010).”

• We noted several issues pertaining to the measurement of wellbeing in this investigation:

1) As it stands, it is not clear why the measures life satisfaction, subjective happiness, and quality of life, were chosen to reflect wellbeing. We recommend that the authors justify this choice by a) providing information on the reliability and validity of the measures chosen in the target population and for the measurement purpose; b) by including additional statistical information on the relationships between these measures (e.g., reporting correlations between these measures).

We have now provided additional information on the choice of the wellbeing score, see lines 142-149:

“This approach follows prior research demonstrating that integrating multiple related indicators captures a broader, more reliable construct of wellbeing ((Bartels, 2015). These measures are reliable and valid in adolescent populations and capture complementary aspects of wellbeing (Savahl et al., 2021). In our sample, correlations between the three wellbeing indicators were moderate to high ($r = 0.39-0.75$), supporting their conceptual relatedness and justifying their combination into a single latent construct.”

2) Insufficient details are provided regarding the construction of the wellbeing latent score (page 7). We recommend that the authors report the results of the factor modelling to construct the single well-being score, including comprehensive model fit statistics.

Thank you for this helpful comment. We have now clarified the construction of the latent wellbeing score in the Methods section and included additional information regarding the factor modeling approach, model fit statistics, and reliability (e.g., lines 170-173). The latent factor was constructed using structural equation modeling with the

regression method, and model fit indices indicated good model fit. Reliability of the latent scores were high.

3) The authors state that the Netherlands Twin Register contains wellbeing measures at ages 14, 16, and 18 (page 6). However, the present investigation only includes wellbeing assessed at ages 14 -16. Why was this the case?

We initially intended to examine data collected at ages 14, 16, and 18 separately. However, due to limited data availability at some of these time points, we decided to focus on the adolescent data collected between ages 14 and 16. We have now added this to the Methods section for clarity, see lines 118-119:

“We used adolescent data collected by the NTR around ages 14 to 16, with a focus on participants with available wellbeing and outcome measures.”

- It would be helpful to provide additional information regarding the choice of specific age categories for adult outcomes (i.e., 20-25 and 25-35).

We thank the reviewer for this comment. The age categories (20–25 and 25–35) were chosen based on data availability and the structure of the longitudinal assessments. We now clarify this in the manuscript (see Methods, lines 183-184):

“Age categories (20–25 and 25–35 years) were selected based on data availability and the timing of follow-up assessments in the cohort.”

Results

- The results section would benefit from greater explanation, e.g. not just stating associations but interpreting them for the reader, too.

We have updated the results section with including interpretation of the results.

- The authors state on page 12 that they calculated the incremental R² as an addition to preregistered analyses to examine the “difference between the R² of the regressions with and without the outcome measure”. However, it is unclear as to which models are compared based on R² statistics to determine this value. We recommend that further details be provided on this.

We thank the reviewer for pointing this out. Incremental R² was calculated for each outcome measure in both the between- and within-family models to assess the proportion of variance in teenage wellbeing explained by each outcome. As now further clarified in the Methods (lines 306-313), incremental R² reflects the difference in explained variance between the full model (including the outcome measure of interest) and the baseline model (including only covariates):

“As an addition to the preregistered analyses, incremental R² was calculated to assess the proportion of variance in adolescence wellbeing explained by each outcome variable for Model 1 and 2. The incremental R² is defined as the difference between the

R² of the full regression model (R², see Supplementary Table S3) and the R² of the corresponding baseline model (R_{null}, see Supplementary Table S3) without the outcome variable. For example, the incremental R² for neuroticism in the between-family analysis was calculated by subtracting the R² of the baseline model (including only covariates) from the R² of the full model predicting wellbeing from neuroticism and covariates.”

Discussion

- The discussion would benefit from a clearer paragraph structure throughout that crystallises key messages for the reader. E.g., a strengths and limitations paragraph would also be useful. The writing could be less descriptive and provide more evaluation and analysis. For instance, in the first paragraph, rather than re-stating technical findings of “sibling fixed effects”, highlighting key findings in relationship to hypotheses as well as implications for theory and practice would be useful.

We have now added subheadings in the discussion to aid readability.

- In the opening paragraph of the discussion, the authors discuss the significant between-family associations found in this study. However, it could be clearer whether the associations described here are specifically those found after controlling for these outcome variables assessed at teenage. This also applies to the description of results in the rest of the discussion section.

Thank you for this important suggestion. We agree that it is crucial to clarify whether associations remain after controlling for baseline adolescent measures. To improve clarity, we have added a sentence in the opening paragraph of the discussion explicitly noting that several between-family associations remained significant after adjusting for baseline outcome levels during adolescence. Furthermore, the detailed sections on between-family and within-family analyses now clearly specify which outcomes were adjusted for adolescent levels and discuss the implications accordingly. We believe these changes improve the transparency of our findings while keeping the discussion coherent.

- Despite evidence for differences in findings between the 20-25 and 25-35 age brackets (e.g., teenage wellbeing only predicts conscientiousness at age 20-25, but not at age 25-35), differences between age groups are not explicitly outlined or discussed in depth in the discussion. It would be helpful to discuss these differences in further detail. Relatedly, there are some inconsistencies in reporting regarding the association between teenage wellbeing and adult hash and marijuana use (this was significant at ages 20-25 and non-significant at ages 25-35).

We have now made this more clear in the Discussion (lines 583-544):

“Beyond sample size considerations, notable differences emerged between the 20–25 and 25–35 age groups. For example, adolescent wellbeing predicted conscientiousness

and cannabis use in the younger group but not in the older group, suggesting potential developmental shifts or changing environmental influences across early adulthood.

Further, it is important to note that ages 14–16 correspond to middle adolescence, and wellbeing and its correlates may vary across different developmental stages (Marquez et al., 2024) highlighting that some influences on wellbeing at this age may have originated earlier in childhood or adolescence. These findings highlight the importance of further exploring age-specific mechanisms in future research”

- On page 20 of the discussion, the authors note that the associations between teenage wellbeing and adult outcomes “underscores the potential for intervention protocols aimed at improving wellbeing at any stage of life.” Given that the present results support an association between teenage wellbeing and adult outcomes, we suggest that the authors’ interpretation is amended to reflect the results better.

We have provided a more suitable interpretation given the focus on adolescence, see lines 502-503:

“...underscores the potential for intervention protocols aimed at improving wellbeing during adolescence, a critical period with long-term implications for adult outcomes.”

- Conscientiousness is incorrectly referred to as consciousness at several points throughout the manuscript, including in the abstract and discussion.

Thank you for the thorough evaluation. We have omitted the typos in the original lines 24,163, and 375.

Reviewer #4:

Thank you for critically reviewing our manuscript.

Response to reviewers

We like to thank the reviewers for evaluating our manuscript again. Below we provide a response to each comment. In the revised manuscript, the changes made are noted in red font colour. We hope you will find the revised manuscript suitable for publication in your esteemed journal.

Reviewer #1 (Remarks to the Author):

I believe the authors have addressed all my concerns, and that the paper now will become a very nice contribution to the literature.

We thank the reviewer for this positive feedback and are happy that the revisions have addressed the earlier concerns.

Reviewer #2 (Remarks to the Author):

Thank you. My concerns have been addressed in the revisions and I do not have anything else to add.

We thank the reviewer for this assessment and are glad that the revisions satisfactorily addressed the concerns raised.

Reviewer #3 (Remarks to the Author):

Thank you for the opportunity to review the revised manuscript, as well as the authors' response letter. We appreciate the authors' efforts to address the points raised in our initial round of review.

The authors now clearly outline the differences between a between-family and within-family design, as well as discuss the strengths and weaknesses of the within-family approach. The manuscript also provides definitions of wellbeing and adolescence. We also note that the authors have made their analysis code publicly available and that the abstract now includes key statistics.

There are several remaining points that we believe would benefit from additional clarification and/or revision:

- We appreciate that the authors now state their hypotheses in the introduction. However, given the wide range of outcomes examined in the present study, it would be helpful if the authors could 1) explicitly comment on why such a broad range of outcome measures were chosen and 2) indicate whether any specific hypotheses were

made for specific outcomes. If no such hypotheses were made due to a lack of prior literature, this could be stated explicitly to aid readers' interpretation.

We thank the reviewer for this suggestion. We have now added text in the Introduction clarifying our rationale. Specifically, we explain that we examined a broad range of outcomes because wellbeing is a multidimensional construct with potential influence on various later life outcomes. We also note that, while we expected the strongest associations with wellbeing, we did not preregister specific hypotheses for other outcome due to the limited prior within-family literature. This has now been explicitly stated (lines 101-108).

“Given the limited prior work on within-family associations between adolescent wellbeing and later outcomes, we examined a broad set of outcome domains, including various sociodemographic, health, and lifestyle measures. This breadth reflects the multidimensional nature of wellbeing, which may influence multiple facets of later life. While we expected the strongest associations with later life outcomes such as wellbeing and self-rated health, we did not preregister specific hypotheses for other outcome domains due to the lack of existing literature. We therefore adopted an exploratory approach, providing a foundation for future, more targeted investigations.”

- We understand that the authors chose not to report the results of the mediation analyses to maintain the readability of the manuscript. However, given that these analyses were preregistered, omitting these analyses is perhaps not optimal from a standpoint of complete reporting and transparency. We suggest that this deviation from analysis plans is explicitly outlined as a deviation from the preregistered analyses in the Methods section.

We agree and appreciate this point. We did not perform the preregistered mediation analyses, as we determined that they would add considerable complexity without strengthening the central conclusions of the paper. We now explicitly acknowledge this deviation from our preregistered plan in the Methods section, see lines 110-114:

“All analyses were preregistered, see <https://osf.io/7dw8z>). Our preregistration included mediation analyses. However, these analyses were not conducted, as we concluded during manuscript preparation that they would add substantial complexity without providing additional clarity on our core research questions. We note this as a deviation from our preregistered plan to ensure transparency. Further minor deviations from the preregistrations are explained in the subsequent subsections.”

In addition, we have added a comment to the preregistration itself explicitly noting that the mediation analyses were not conducted and explaining the rationale for this deviation to ensure full transparency.

- We appreciate that there were varying sample sizes per outcome due to differences in the measures included in each survey, and that most participants were not surveyed for every item. However, looking at Table 1, sample sizes seem to vary considerably across

outcome variables, and some appear quite small (e.g., sleep duration at age 25-35: N = 29). We suggest that the authors comment on this variation and consider whether analyses for certain outcomes may have been underpowered.

We thank the reviewer for highlighting this. We have now added text in the Methods to acknowledge the variation in sample sizes across outcomes (lines 196-199). In addition, in the Discussion we note that some results, particularly those based on small subsamples (e.g., sleep duration at ages 25–35, N = 29), may be underpowered and should be interpreted with caution (lines 586-589).

*“Due to differences in the measures included across survey waves, sample sizes varied considerably across outcomes (see **Table 1**). This means that most participants did not contribute data to every analysis.”*

and

“Importantly, variation in outcome-specific sample sizes likely reduced statistical power for certain analyses. For example, sleep duration at ages 25–35 was based on a relatively small subsample (n = 29). Results based on such limited sample sizes should be interpreted with caution.”

- When the authors state that the regression method was used to create the factor score, do they mean that a confirmatory factor analysis (CFA) was first conducted to model the latent construct, followed by using regression to compute individual scores? Clarifying this process and including a path diagram would help readers better understand the measurement approach.

We appreciate this request for clarification. Adolescent wellbeing was modelled as a latent construct using CFA. Individual scores were then computed using the regression method. We have clarified this procedure in the Methods section (lines 170-177). In addition, we now provide a schematic path diagram of the measurement model in the Supplementary Materials (Supplementary Figure S1).

“Participants with at least two out of three wellbeing measures (life satisfaction, subjective happiness, quality of life) were included. The three measures were standardized prior to analysis. Adolescent wellbeing was then modeled as a latent construct using confirmatory factor analysis (CFA) with these three indicators. A schematic path diagram of the measurement model is provided in Supplementary Figure S1. Factor scores for each participant were computed using the regression method, which ensures that the covariance matrix of the estimated scores matches the model-based latent covariance structure (Croon, 2002; Devlieger & Rosseel, 2017).”

- We thank the authors for providing additional information regarding the incremental R². However, it remains unclear how an R² can be calculated "without the outcome variable".

We thank the reviewer for pointing out this ambiguity. We have now rephrased our description for clarity, see lines 315-320. Specifically, incremental R² is defined as the

difference between two nested models: a baseline model including only covariates, and a full model including covariates plus the predictor of interest.

*“Incremental R^2 was defined as the difference between two nested models: (1) a baseline model including only covariates, and (2) a full model including covariates plus the predictor of interest. For example, the incremental R^2 for neuroticism in the between-family analysis was obtained by subtracting the R^2 of the baseline model (covariates only) from the R^2 of the full model predicting wellbeing from neuroticism and covariates (see **Supplementary Table S3**).”*

- We appreciate that the authors have now commented on the differing findings between adult age groups. However, it would be helpful if the authors could elaborate further on what is meant by “potential developmental shifts or changing environmental influences across early adulthood” and how this might help to interpret the observed results.

We agree and have expanded our discussion of this point. We now highlight that early adulthood (ages 20–25) is characterized by major life transitions such as higher education, entering the workforce, forming new relationships, and experimenting with health behaviors. Later young adulthood (ages 25–35), in contrast, is typically marked by greater stability in work, family, and lifestyle patterns, as well as increased health and caregiving responsibilities. We suggest that these developmental and environmental shifts may help to explain the age-specific findings. This elaboration has been added to the Discussion (lines 595-602).

“One possible explanation is that early adulthood (age 20–25) is a period characterized by major life transitions such as entering higher education, establishing independence from parents, forming new peer and romantic relationships, and experimenting with health behaviors. By contrast, later young adulthood (age 25–35) often involves greater stability in work, family, and lifestyle patterns, as well as increased health and caregiving responsibilities (Marquez et al., 2024). These developmental and environmental shifts may alter the extent to which adolescent wellbeing predicts later outcomes, potentially explaining the observed age-dependent differences.”

Reviewer #4 (Remarks to the Author):

Thank you for reviewing our manuscript and the constructive feedback.

Response to reviewers

We thank the reviewers for re-evaluating our manuscript. Below, we provide a response to each comment. In the revised manuscript, changes are indicated in red font. We hope that the revised version is now suitable for publication.

Reviewer #3 (Remarks to the Author):

We thank the authors for their thoughtful response to our feedback. Our concerns have been addressed.

This is a minor point, but we recommend aligning the CFA path diagram with field conventions, e.g., by showing factor loadings and variances.

We thank the reviewers for this suggestion. We have updated Supplementary Figure S1 to align with standard CFA conventions by explicitly displaying standardized factor loadings and variances.

Reviewer #4 (Remarks to the Author):

We thank the reviewer for their time and contribution to the evaluation of our manuscript.